# MODIS and PROBA-V NDVI Products Differ when Compared with Observations from Phenological Towers at Four Tropical Dry Forests in the Americas

**J. Antonio Guzmán Q. [1]**, **G. Arturo Sanchez-Azofeifa [1,*]** and **Mário M. Espírito-Santo [2]**

[1] Centre for Earth Observation Sciences, Department of Earth and Atmospheric Sciences, University of Alberta, Edmonton, AB T6G 2E3, Canada; guzmnque@ualberta.ca

[2] Department of Biological and Health Sciences, State University of Montes Claros—Unimontes, Montes Claros, MG 39401-089, Brazil; mario.marcos@unimontes.br

**\*** Correspondence: arturo.sanchez@ualberta.ca; Tel.: +01-780-492-1822

**Abstract:** The Normalized Difference Vegetation Index (NDVI) is widely used to monitor vegetation phenology and productivity around the world. Over the last few decades, phenology monitoring at large scales has been possible due to the information and metrics derived from satellite sensors such as the Moderate Resolution Imaging Spectroradiometer (MODIS) or the Project for On-Board Autonomy–Vegetation (PROBA-V). However, due to their temporal and spatial resolution, adequate ground comparison is lacking. In this paper, we analyze how NDVI products from MODIS (Aqua and Terra) and PROBA-V predict vegetation phenology when compared with near-surface observations. We conduct this comparison at four tropical dry forests (TDFs) in the Americas. We undertake this study by comparing the following: (i) Dissimilarities of the standardized NDVI (NDVI$_S$) using dynamic time warping, (ii) the differences of daily NDVI$_S$ between seasons and ENSO months using generalized linear models, and (iii) phenometrics derived from NDVI time series. Overall, our results suggest that NDVI$_S$ from satellite observations present DTW distances (dissimilarities) between 2.98 and 46.57 (18.91 ± 12.31) when compared with near-surface observations. Furthermore, NDVI$_S$ comparisons reveal that overall differences between satellite and near-surface observations are close to zero, but this tends to differ between seasons or when El Nino Southern Oscillation (ENSO) is present. Phenometrics comparisons show that metrics derived from satellite observations such as green-up, maturity, and start and end of the wet season strongly correlate with those from near-surface observations. In contrast, phenometrics that describe the day of the highest or lowest NDVI tend to be inconsistent with those from near-surface observations. All findings were observed independently of the NDVI source. Our results suggest that satellite-based NDVI products tend to be inconsistent descriptors of vegetation events on tropical deciduous forests in comparison with near-surface observations. These results reinforce the idea that satellite-based NDVI products should be used and interpreted with great caution and only in ecosystems with well-established knowledge of their vegetation phenology.

**Keywords:** phenology; tropical dry forest; vegetation index; ENSO; satellite observations

## 1. Introduction

Monitoring vegetation phenology is fundamental to the inference of biochemical cycles and their relationship to environmental conditions and stresses. Currently, vegetation phenology is considered by Pereira et al. [1] to be part of a suite of Essential Biodiversity Variables since it helps to know the status and trends of processes in the ecosystems (e.g., green-up, flowering, or herbivory) [2]. Traditionally,

vegetation phenology has been monitored at large scale using earth observation approaches and their derived vegetation indices [3,4]. Current satellite missions, such as Moderate Resolution Imaging Spectroradiometer (MODIS) from the National Aeronautics and Space Administration or Project for On-Board Autonomy–Vegetation (PROBA-V) from the European Spatial Agency provide vegetation indices with the potential to describe the vegetation dynamics on a given landscape at reasonable spatial and temporal resolutions.

Products such as the Normalized Difference Vegetation Index (NDVI) have been widely used as indicators of vegetation phenology over the last few decades at the local, regional, and global levels [4–7]. In general, this index, derived from the red:near-infrared ratio (NDVI = (NIR − RED)/(NIR + RED)) describes how the red light is absorbed by the vegetation in comparison with the near-infrared light that tends to be scattered when encountering with vegetation [8]. The relationship between NDVI with parameters such as vegetation health, productivity, or the fraction of absorbed photosynthetic active radiation (*f*PAR) is well documented [9–12]. However, because of its nature, satellite-based NDVI has limitations regarding its validation. For example, variations in the spectral, spatial, and radiometric features between sensors may have an influence in the variability of NDVI [13]. Likewise, the use of NDVI products must be carefully considered since these observations could be affected by a large number of factors (e.g., variations in solar zenith and viewing angles, atmospheric conditions, topography, cloud cover, surface reflectance bidirectional effects, and leaf area index, among many) [14–16].

Although the study of the vegetation phenology in the temperate regions using vegetation indices is well documented, this kind of study in tropical dry forests (TDFs) is limited [17,18]. At a regional scale, for example, the application of vegetation indices to study the TDF phenology has been used to characterize vegetation types [19], succession states [18], and the ecosystem productivity [20,21]. At a continental or global scale, however, these types of applications are even scarcer and tend to be focused on characterizing different ecosystems [6,22,23]. The importance of studying the TDF phenology lies in that this ecosystem has been considered one of the most threatened due to its intensive disturbance, seasonality, and lack of protection [24–26]. The phenology of these regions is mainly driven by the temporal rainfall distribution, where plants tend to produce or shed leaves as a response to changes in soil moisture. Due to their fragmentation and disturbance, the TDF tend to be associated with forest patches of different ages [27], where the species composition and their structure depends on the successional stages of these patches [28,29]. These elements of composition and structure of the forest, which vary with the successional stages, affect the vegetation indices captured by satellites and their phenology [18]. As such, a good understanding of TDF phenological expressions is essential to unravelling key ecological processes affecting these ecosystems' functioning as well as their resilience to extreme meteorological events such as El Nino Southern Oscillation (ENSO).

In this paper, we analyze how well NDVI satellite products from MODIS (Aqua and Terra) and PROBA-V are associated with NDVI derived from near-surface observations at four sites in the TDF located across the Americas. This work expands on previous regional studies that compared NDVI satellite products to optical phenology towers [7,18,30] by exploring three questions, as follows: (i) How similar are satellite NDVI time series to near-surface NDVI time series? (ii) How variable are the differences in standardized NDVI values between satellite and near-surface observations during wet and dry seasons or ENSO months? Finally, (iii) how comparable are phenological events derived from satellite observations to those extracted from in situ observations? In this paper, our main goal is to explore sources of variability in these products in comparisons with high temporal resolution in situ measurements. We further aim to provide considerations as to how these products may fail to detect TDF vegetation phenology.

## 2. Materials and Methods

### 2.1. Study Sites

We conducted this study at four TDFs in the Americas (Figure 1). These sites are home to long term phenology and carbon flux studies by Tropi-Dry (www.tropi-dry.org). These sites represent four different TDF communities with particular characteristics of temperature, precipitation, and length of dry season (Table 1). The first site is located at the Chamela Biological Station (CBS), Jalisco, Mexico. This site presents an undisturbed TDF with drought-deciduous vegetation [31]. The phenological and flux tower height is close to 10.5 m. The second site is located at the Santa Rosa National Park–Environmental Monitoring Super Site (SRNP-EMSS), Costa Rica. This site presents a mosaic of forest patches with different ages [27] resulting from different land-use histories associated with anthropogenic fires, intense deforestation, and clearing lands for pasture [28,32]. The tower height at this site is close to 18.0 m. The other two sites are located at the Lagoa do Cajueiro State Park (LC-SP) and Parque Estadual da Mata Seca–Environmental Monitoring Super Site (PEMS-EMSS). These two sites present a mosaic of forest covers at different stages of secondary succession, which is surrounded by an agricultural matrix [33]. The LC-SP tower height is close to 15 m. At the PEMS-EMSS there are four towers located along a successional gradient, as follows: (i) Pastures (canopy height = 3 m), early (5–20 years, tower height = 8 m), intermediate (20–50 years, tower height = 15 m), and late forest (>50 years, tower height = 22 m) (See Rankine et al. [18] for a detailed description of these forest sites). The height of these towers varies between sites; however, these maintain a distance close to 3–6 m above the forest canopy.

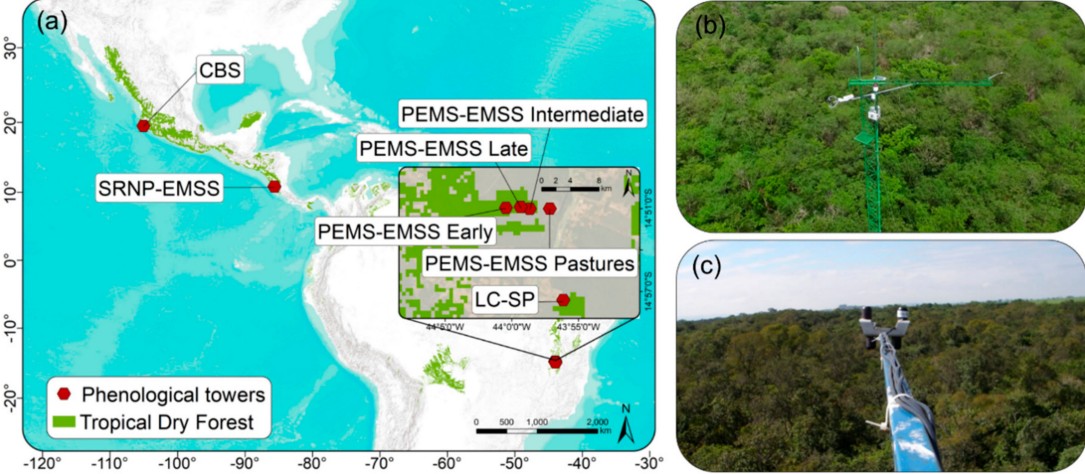

**Figure 1.** Location map of the optical phenological towers at four tropical dry forests distributed along the Americas (**a**) and images of the towers (**b**) and sensors used (**c**). Chamela Biological Station (CBS), Santa Rosa National Park Environmental Monitoring Super Site (SRNP-EMSS), Lagoa do Cajueiro State Park (LC-SP), and Parque Estadual da Mata Seca–Environmental Monitoring Super Site (PEMS-EMSS). Observations at the PEMS-EMSS come from four towers located in sites with different successional stages. The tropical dry forest distribution is based on Portillo-Quintero and Sánchez-Azofeifa [26].

**Table 1.** Main characteristic of the sites where the optical phenological towers are located at four tropical dry forests along the Americas. Mean annual temperature (MAT), average annual precipitation (AAP).

| Country | Site | Acronym | Type of Cover | Tower Height (m) | Altitude (m a.s.l.) | MAT (°C) | AAP (mm) | Dry Season | Range of Observations (Day-Month-Year) Start | Range of Observations (Day-Month-Year) End |
|---|---|---|---|---|---|---|---|---|---|---|
| Mexico | Chamela Biological Station | CBS | Secondary forest | 10.5 | 75 | 25.7 | 790 | November–April | 22-03-2008 | 27-10-2014 |
| Costa Rica | Santa Rosa National Park [1] | SRNP-EMSS | Secondary forest | 18.0 | 290 | 25.0 | 1720 | December–May | 23-02-2013 | 29-03-2016 |
| Brazil | Lagoa do Cajueiro State Park | LC-SP | Secondary forest | 15 | 471 | 24.4 | 871 | April–October | 02-06-2012 | 14-11-2014 |
| Brazil | Parque Estadual da Mata Seca [1] | PEMS-EMSS | Pastures | 3 | 455 | 24.4 | 900 | April–October | 21-04-2012 | 17-07-2015 |
| | | | Early forest | 8 | 469 | | | | 20-06-2009 | 30-10-2013 |
| | | | Intermediate forest | 15 | 486 | | | | 04-10-2007 | 07-05-2011 |
| | | | Late forest | 22 | 486 | | | | 02-11-2010 | 06-04-2016 |

[1] CEOS Environmental Monitoring Super Site.

### 2.2. Phenological Towers Observations

In all the tower sites, canopy observations were conducted using a set of two hemispherical silicon pyranometers (Apogee SP-110, Utah, USA) and two quantum sensors (Apogee SQ-110, Utah, USA). These sensors captured the incident and reflected shortwave solar radiation (360–1120 nm) and photosynthetically active radiation (410–655 nm) at the top of the tower, respectively. Both sensors have a field of view of 180° and a directional response of ± 5% at 75° zenith angle. These sensors were mounted at the end of a three-meter boom off the top of the towers to avoid observations from the tower structure (Figure 1c). At their distance above the forest canopy and with their field of view, these sensors allow capturing an effective circular canopy footprint of an area close to 1.76 ha. Both pyranometers and quantum sensors were synchronized in each tower to sample canopy observations every 30 s, and then averaged and recorded at 15 min intervals. Sensors at all four sites started to record data at different dates; the range of dates used for this study is provided in Table 1. From each synchronized tower NDVI was estimated following Wilson and Meyers [34]. The NDVI values obtained were filtered by time between 10:00 and 14:00 and photon fluxes above 1000 $\mu$mol m$^{-2}$ s$^{-1}$ were removed to avoid high solar zenith angles effects and possible cloud interferences. The filtered values were averaged per day and then were processed following the procedures described below in the Data Preprocessing section.

### 2.3. MODIS Observations

Level three NDVI products from MODIS satellite data of collection six were used from both Terra (MOD13Q1) and Aqua (MYD13Q1) sensors [35,36]. These products were extracted using the Oak Ridge National Laboratory DAAC MODIS land process subset tool [37]. Specifically, the NDVI values are level-3 products of 16-day time series observations with a spatial resolution of 250 m. These products were created using a compositing technique that extracts the best quality pixel from a group of observations for a given 16-day period. The best quality pixel is considered as a cloud-free and nadir view pixel with no residual atmospheric contamination [38]. These products tend to be used for characterizing biophysical properties and processes on land surfaces, such as primary productivity and land cover change [37]. A pixel of these products was extracted for each tower using the ground coordinates with a date range between 2007 and 2018. NDVI products extracted for both sensors were then merged by day to produce an 8-day time series, which was cut later to match with the temporal range of each near-surface observation. The resulting NDVI time series were processed to extract daily observations following the methods described in the Data Analysis section. These products were selected over the daily-raw MODIS surface reflectance because they are more readily accessed and commonly used by the scientific community that works with vegetation phenology and temporal anomalies [7,18].

### 2.4. PROBA-V Observations

Level-3 NDVI product of PROBA-V was also used in this study. Similar to MODIS products, this product is atmospherically corrected and quality controlled to be masked for clouds and aerosols. The PROBA-V NDVI product was extracted from the Copernicus Global Land Service (https://land.copernicus.eu/global/products/ndvi). Specifically, this product is composed of global NetCDF4 files of a 10-day time series of NDVI observations with a spatial resolution of 333 m. This product is a synthesis of observations based on the selection of best measurements, which is the maximum NDVI observation in a given 10-day period having a zenith angle smaller than 40° or a sun zenith angle smaller than 60° [39]. In general, this method offers pixels with the least atmospheric attenuation and viewing geometry effects. From these global files, a pixel was also extracted per tower site with a date range between 2014 and 2018. The resulting time series were cut to match the temporal range of each near-surface observation. Due to the temporal unavailability of the PROBA-V product at the same time-frame, the early and intermediate phenology towers at the PEMS-EMSS were not used when

compared against this product. The resulting NDVI time series were also processed to extract daily observations following the methods described below.

## 2.5. Data Preprocessing

The resulting NDVI data from near-surface, MODIS, and PROBA-V observations were processed for a series of steps to fill gaps and to produce a daily smoothed time series. These steps were performed to correct sporadic sensor failures from the phenological towers, as well as to compensate for the presence of clouds or the atmospheric variability of MODIS and PROBA-V observations. Specifically, the phenology tower observations were the first gap filled and then smoothed. On the other hand, MODIS and PROBA-V observations were first smoothed with their original frequency of observation (8 days and 10 days, respectively) and then gap filled and smoothed to daily observations. The gap filling method consists in a linear missing value imputation using the function 'na.interpolation' of the *imputeTS* package [40] of R [41], while the smoothing method consists in the application of a Savitzky–Golay polynomial filter with a moving window of 25 observations using the function 'sgolayfilt' of the *signal* package of R [42]. In general, the Savitzky–Golay filter was used instead of other methods such as the BISE algorithm or Fourier-based fitting because it has been extensively used in other studies and proved to be a proper method to produce high-quality NDVI time series [43–45]. Preprocessed data of daily NDVI observations from phenological towers are available at https://doi.org/10.7910/DVN/BDCJNP. Likewise, Figures of unprocessed and preprocessed daily observations from satellite products are available in Supplementary Materials (Figures S1 and S2).

Since the NDVI values are by definition sensor-dependent [13] and the magnitude of variation of NDVI depends on how the light interacts with the surfaces [46], as well as the preprocessing of these products (e.g., aerosols or atmospheric corrections), we performed a standardization of the NDVI from both the satellite and the near-surface observations. Precisely, the variability of each NDVI time series was standardized using the following formula: (observation − minimum)/(maximum − minimum). This was applied to the entire multi-annual time series for each sensor and pixel. On the near-surface time series, this formula was applied twice per case of match of observations with MODIS and PROBA-V. With this method, the standardized NDVI ($NDVI_S$) values were transformed to a range between 0 and 1, avoiding the effects of the variation in amplitude between sensors and allowing comparisons between NDVI trends. Calculations of the $NDVI_S$ have also been performed by studies to validate MODIS products using landscape phenology indices from plot-level observations [47] or to study temporal anomalies [48].

## 2.6. Comparisons between Phenological Towers and Satellite Observations

The daily NDVI and $NDVI_S$ time series of the satellite sensors were compared with the optical phenology towers to estimate how variable were the satellite observations from the near-surface observations. These comparisons were performed assuming that (i) the NDVI observations from the optical towers are not affected by canopy-air-sensor medium, and (ii) the area covered by the optical tower sensor is the same as that within a pixel of the satellite-derived NDVI product. In the previous assumption, it is difficult to ensure that the near-surface sensor footprint matches with the center of the pixels from the satellite products; however, due to the homogeneity of the sites near the towers, we may assume that the representativeness of the covers is true. Under these assumptions, the satellite and near-surface observations were compared at three levels, as follows: (i) Overall dissimilarities between $NDVI_S$ time series, (ii) daily differences of $NDVI_S$, and (iii) comparing phenometrics derived from NDVI observations.

### 2.6.1. $NDVI_S$ Dissimilarities between Near-Surface and Satellite Observations

Similarity or temporal distortions between near-surface and satellite observations were estimated using a dynamic time warping (DTW) analysis. This analysis was applied by comparing the daily $NDVI_S$ time series of satellite and near-surface sensors at the same time intervals. In general, the DTW

analysis enables one to describe the quality of global alignment between two time series, looking at the optimal match between sequences. Within this analysis, first a Euclidian distance matrix is estimated between all pairs of points of the $NDVI_S$ sequences. Then a warping path from the beginning to the end of the sequences is computed. The aim of this path is to find the minimum accumulative cost of distance between consecutive observations. The optimal match between sequences, represented by the minimum accumulative distance, describes the overall cost of this alignment and, at the same time, the dissimilarity between time series [49]. The accumulative distance has no units since it comes from comparisons between dimensionless sequences. In general, distance values close to zero describe pairs of time series with high similarity of the phenological patterns. This analysis was conducted using the R package 'dtw' [50] assuming that the relationship between the time series has a symmetric step pattern.

### 2.6.2. Comparisons of NDVI between Satellite and Tower Observations

Differences of $NDVI_S$ ($NDVI_{DS}$) between satellite and near-surface observations per day were performed to compare how variable were the satellite observations in comparison with those from optical tower. In essence, $NDVI_{DS}$ values close to 0 represent higher similarities of the phenological trends between satellite and tower observations, while $NDVI_{DS}$ values higher or lower than 0 represent an over- or under-estimation of the vegetation phenology derived from satellite products in comparison with optical phenology towers.

Density plots were performed using the $NDVI_{DS}$ values to represent how far were the distribution of values from 0. In addition, density plots of $NDVI_{DS}$ values from different seasons and ENSO months were performed to compare how the $NDVI_{DS}$ distribution may be affected for meteorological events. For this, the extraction of dates of the start and end of the wet season from phenological towers (See the Section 2.6.3) was used to describe when the $NDVI_{DS}$ values fall in wet or dry seasons. Likewise, historical ENSO events calculated by the Climate Prediction Center (National Weather Service, NOAA, www.nws.noaa.gov), based on a monthly threshold of ± 0.5 °C of the Oceanic Niño Index, were used to describe when the $NDVI_{DS}$ values fell in El Niño ($<-0.05$), La Niña ($>0.05$), or normal ($-0.05$–$0.05$) months. In all the cases, density plots were performed using kernel density based on a Gaussian distribution. Moreover, statistical comparisons of $NDVI_{DS}$ between seasons and ENSO months were conducted through generalized least squares linear models, using $NDVI_{DS}$ as a dependent factor and seasons or ENSO months as independent factors. Given the nature of the $NDVI_{DS}$ data, temporal autocorrelations were considered in the models using a first-order continuous autoregressive process to avoid correlations between consecutive values. Density plots were performed using the function 'density', while the generalized least squares models were conducted using the function 'gls' of the package *nlme* [51] of R.

### 2.6.3. Extraction of the Phenometrics

Seasonal phenophases from NDVI observations were estimated to analyze the temporal differences of the phenometrics between optical phenology towers and satellite-based products. These phenometrics were manually estimated following the method described by Tan et al. [52]. In this sense, the first and third derivatives of the filter performed above on daily time series were extracted in order to estimate phenology dates. Specifically, the annual local maximum and minimum of the first derivative were used to determine the day of the year (DOY) of the start and end of the wet season (SWS and EWS). Likewise, the annual local maximum and minimum of the third derivative curve represented the green-up, maximum increasing slope, maturity, beginning of the senescence, maximum decreasing slope, and dormancy. In this study, only the phenometrics of green-up and maturity from the third derivative were used because the other local maximum and minimum of the derivative did not present pronounced peaks or depressions to extract them with certainty. Additionally, the annual local maximum and minimum of the NDVI time series were also used to describe the DOY of the highest and lowest NDVI value (DHV and DLV). The six phenometrics from satellite sensors were regressed against

near-surface observations using a standardized major axis (SMA). This is an appropriate analysis when the slope is the main statistic to estimate, since it minimizes the straight-line distance of points from the line [53]. With this analysis we also tested if the satellite observations (predicted variables) present a 1:1 relationship with near-surface observations (observed variable). This analysis was computed using the 'sma' function of the *smart* package in R [54].

## 3. Results

### 3.1. NDVI Time Series and Standardized NDVI Dissimilarities

The processed time series of NDVI from near-surface and satellite-based observations reveal that they tend to differ in their magnitude of NDVI (Figure 2). Specifically, it seems that the differences in the NDVI magnitude between near-surface and satellite observations tend to be more pronounced at higher latitudes (e.g., CBS, LC-SP, or PEMS-EMSS; Figure 2a,c,e,f,g) than close to the Equator (e.g., SRNP-EMSS; Figure 2b). The exception of the latter occurs in the Pasture site of PEMS-EMSS, where the NDVI of near-surface and satellite observations seem to be similar (Figure 2d). On the other hand, comparisons using DTW of the NDVI time series between optical phenology towers and satellite-based observations reveal distances in a range of 2.98 to 46.57, regardless of the satellite sensor (Table 2). Specifically, near-surface–MODIS comparisons show distance values between 12.87 and 46.57 (23.91 ± 10.63), while near-surface–PROBA-V comparisons present values between 2.98 and 32.96 (11.91 ± 11.01). The lowest similarities of NDVI time series between near-surface and satellite observations seem to be at CBS and PEMS-EMSS Late successional sites.

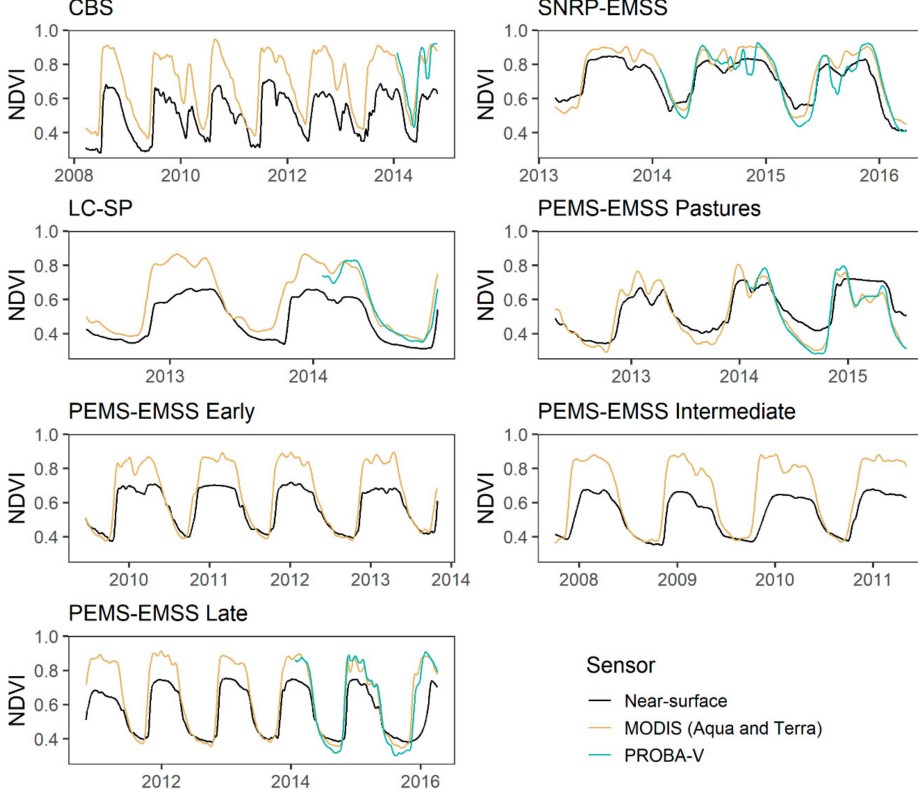

**Figure 2.** Time series of Normalized Difference Vegetation Index (NDVI) derived from phenology towers, MODIS (Moderate Resolution Imaging Spectroradiometer), and PROBA-V (Project for On-Board Autonomy–Vegetation) observations at four different tropical dry forest locations. Observations at the Parque Estadual da Mata Seca (PEMS-EMSS) come from four sites with different successional stages.

**Table 2.** Minimum global distance computed using the dynamic time warp algorithm to compare the differences of standardized NDVI time series between satellite and near-surface observations at four tropical dry forests. Observations at the Parque Estadual da Mata Seca (PEMS-EMSS) come from four successional stages.

| Site | DTW Distance | |
|---|---|---|
| | **Near Observations-MODIS** | **Near Observations-PROBA-V** |
| CBS | 46.57 | 3.54 |
| SRNP-EMSS | 21.18 | 32.96 |
| LC-SP | 12.87 | 2.98 |
| PEMS-EMSS Pastures | 31.12 | 11.92 |
| PEMS-EMSS Early | 16.93 | — |
| PEMS-EMSS Intermediate | 17.54 | — |
| PEMS-EMSS Late | 21.18 | 8.14 |
| Mean (± SD) | 23.91(10.63) | 11.91 (11.01) |

*3.2. Differences between Satellite and Near-Surface Observations of Standardized NDVI*

Overall, our results reveal that the density distributions of the $NDVI_{DS}$ between near-surface observations and MODIS tend to be close to 0 (Figure 3a,d,g,j,m,p,s). This trend does not seem to follow in some sites when the $NDVI_{DS}$ between optical towers and PROBA-V are compared (Figure 4a,d,h,k,n). Likewise, the density distribution appears to be more contrasting and different from 0 when these are compared between seasons or ENSO months. Specifically, results comparing near-surface observations and MODIS show that the $NDVI_{DS}$ tend to be higher in five sites (CBS, PEMS-EMSS Pastures, Early, and Intermediate) during the wet season but not during the dry season (Table 3, Figure 3b,k,n,q). The exception of this trend is present at LC-SP and PEMS-EMSS Late succession sites (Figure 3h,t), where the $NDVI_{DS}$ comparisons seems to have an inverse pattern to the one above. $NDVI_{DS}$ comparisons at SRNP-EMMS seem not to be affected by the seasons (Figure 3e). Results comparing near-surface observations and PROBA-V reveal that the $NDVI_{DS}$ tend to differ between seasons at the PEMS-EMSS Pastures and Late succession sites (Figure 4l,o and Table 3), but not at CBS, SRNP-EMSS, and LC-SP (Figure 4b,f,i and Table 3). Comparing among ENSO months, the results showed that the $NDVI_{DS}$ derived from near-surface and MODIS observations differ in CBS, PEMS-EMSS Pastures, and PEMS-EMSS Late succession (Figure 3c,l,u, Table 3). Similar to above but comparing near-surface and PROVA-V observations, the $NDVI_{DS}$ differed in SRNP-EMMS and PEMS-EMSS Pastures during ENSO months (Figure 4g,m, Table 3). Overall, although some comparisons of $NDVI_{DS}$ between near-surface and satellite sensors reveal differences between seasons and ENSO months, there is not a clear trend of higher or lower $NDVI_{DS}$ values during the wet or dry season, or in a given ENSO month.

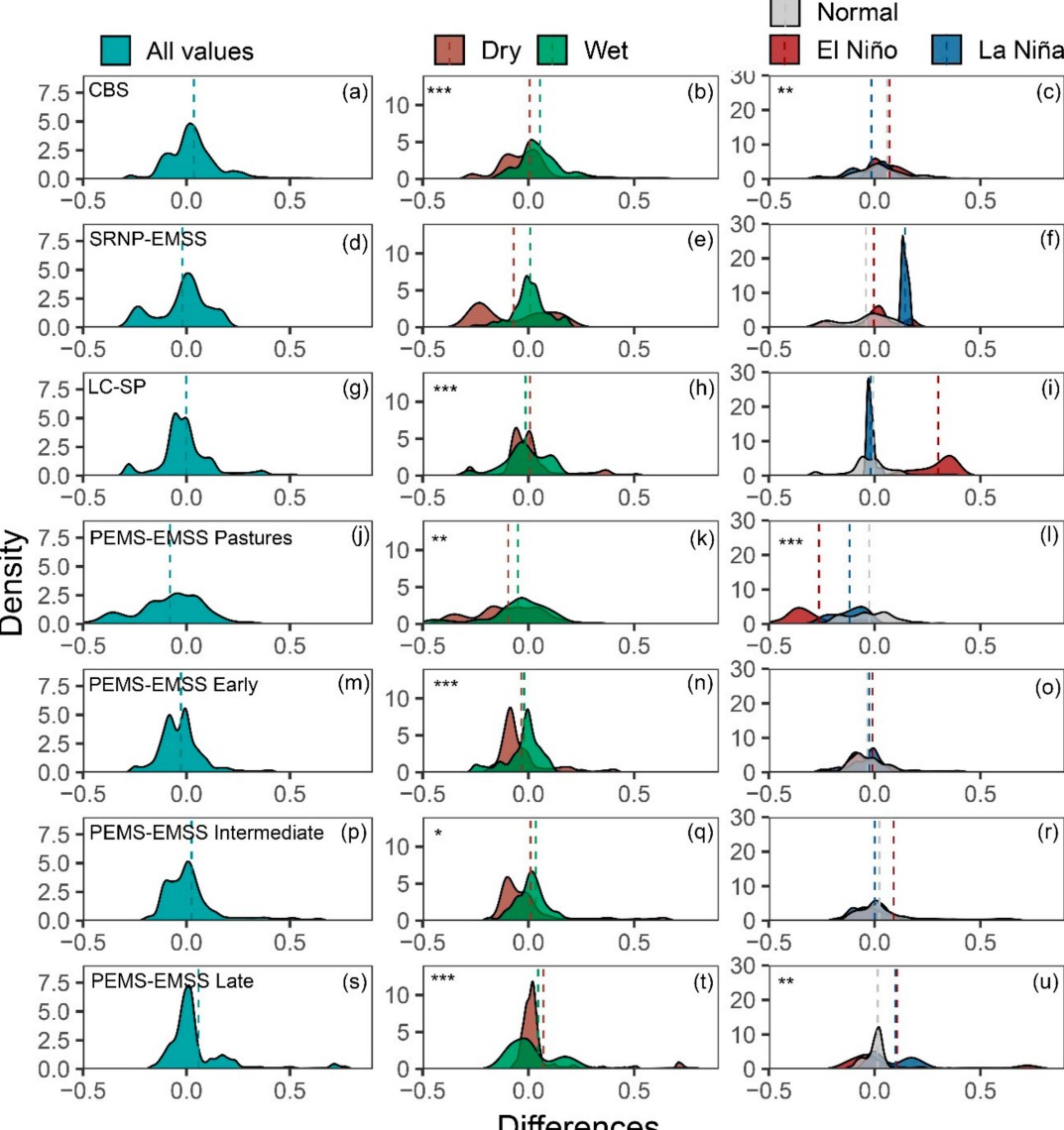

**Figure 3.** Density distribution of comparing standardized NDVI derived from near-surface and MODIS observations (Aqua and Terra) in different seasons (wet and dry season) and ENSO events at four tropical dry forests. Density curves were created using a Gaussian distribution with a bandwidth of 1. Vertical dotted lines represent the mean value of the overall errors per category. Asterisks describe the significant differences of generalized least squares models between seasons or ENSO months (See Table 4 for detailed representation), as follows: * $p < 0.05$; ** $p < 0.01$; *** $p < 0.001$.

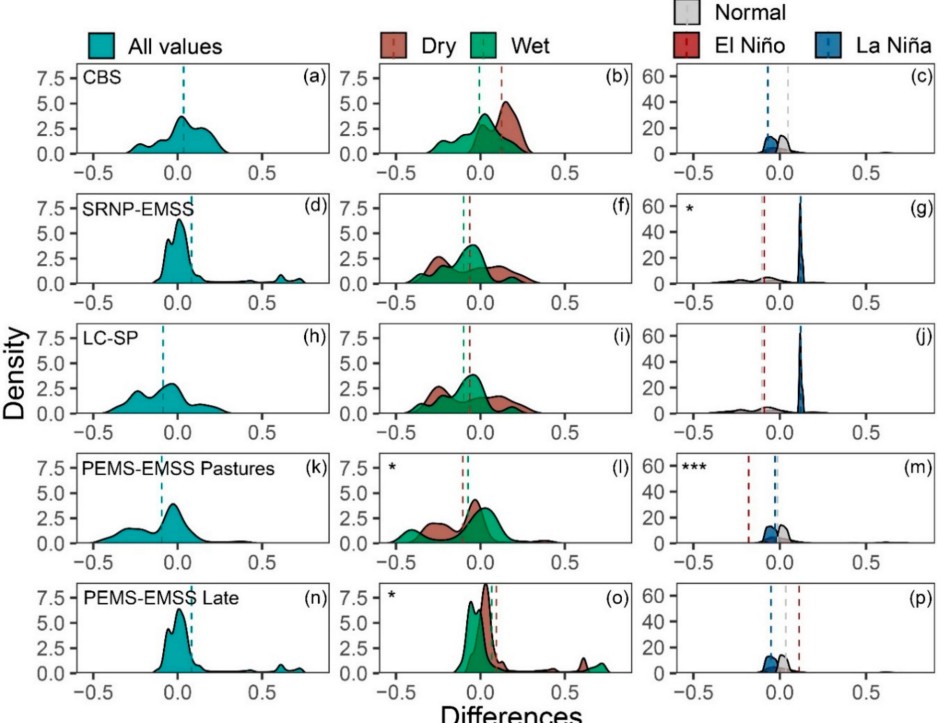

**Figure 4.** Density distribution of the differences comparing standardized NDVI derived from near-surface and PROBA-V observations in different seasons (wet and dry season) and ENSO events at four tropical dry forests. Density curves were created using a Gaussian distribution with a bandwidth of 5. Vertical dotted lines represent the mean value of the overall errors per category. Asterisks indicate the significant differences of generalized least squares models between seasons or ENSO months (See Table 3 for detailed representation), as follows: * $p < 0.05$; ** $p < 0.01$; *** $p < 0.001$.

**Table 3.** Results of the linear model using generalized least squares comparing the temporal and seasonal or ENSO effects (independent factors) on the variation of the differences of standardized NDVI between near-surface and satellite observations (MODIS and PROBA-V) at four tropical dry forests. Observations at the Parque Estadual da Mata Seca (PEMS-EMSS) come from four successional stages.

| | MODIS | | | | PROBA-V | | | |
|---|---|---|---|---|---|---|---|---|
| **Site** | **Season** | | **ENSO** | | **Season** | | **ENSO** | |
| | *F* | *p*-Value | *F* | *p*-Value | *F* | *p*-Value | *F* | *p*-Value |
| CBS | 22.34 | <0.001 | 6.46 | <0.001 | 1.60 | 0.21 | 0.75 | 0.39 |
| SRNP-EMSS | 2.83 | 0.09 | 1.36 | 0.26 | 0.49 | 0.49 | 3.90 | 0.02 |
| LC-SP | 16.63 | <0.001 | 1.10 | 0.33 | 0.10 | 0.75 | 0.33 | 0.72 |
| PEMS-EMSS Pastures | 8.65 | 0.01 | 9.39 | <0.001 | 5.06 | 0.02 | 7.16 | <0.001 |
| PEMS-EMSS Early | 28.72 | <0.001 | 0.08 | 0.92 | — | — | — | — |
| PEMS-EMSS Intermediate | 4.15 | 0.04 | 0.09 | 0.92 | — | — | — | — |
| PEMS-EMSS Late | 24.58 | <0.001 | 5.13 | 0.01 | 4.56 | 0.03 | 1.57 | 0.21 |

### 3.3. Comparisons of Phenometrics Derived from Satellite and Near-Surface Observations

The SMA regressions reveal that some phenometrics derived from near-surface observations tend to have a strong 1:1 relationship with those derived from satellite observations (Figure 5). Specifically, the SMS regressions indicate that the DOY of the phenometrics derived from both satellites, such as the SWS, EWS, green-up, and maturity showed relationships with $R^2$ higher than 0.82 and slopes close to 1. On the other hand, the DHV and DLV phenometrics extracted from satellite observations seem to be inconsistent with those derived from near-surface observations (Figure 5). For example,

the relationship of DHV between MODIS and near-surface sensors showed a $R^2 = 0.45$, while the relationship of DLV between PROBA-V and near-surface sensors had a $R^2 = 0.34$. Despite the low association between sensors in these phenometrics, the test which compares the slopes suggests that all slopes from the SMA regressions do not differ from 1 (Table 4).

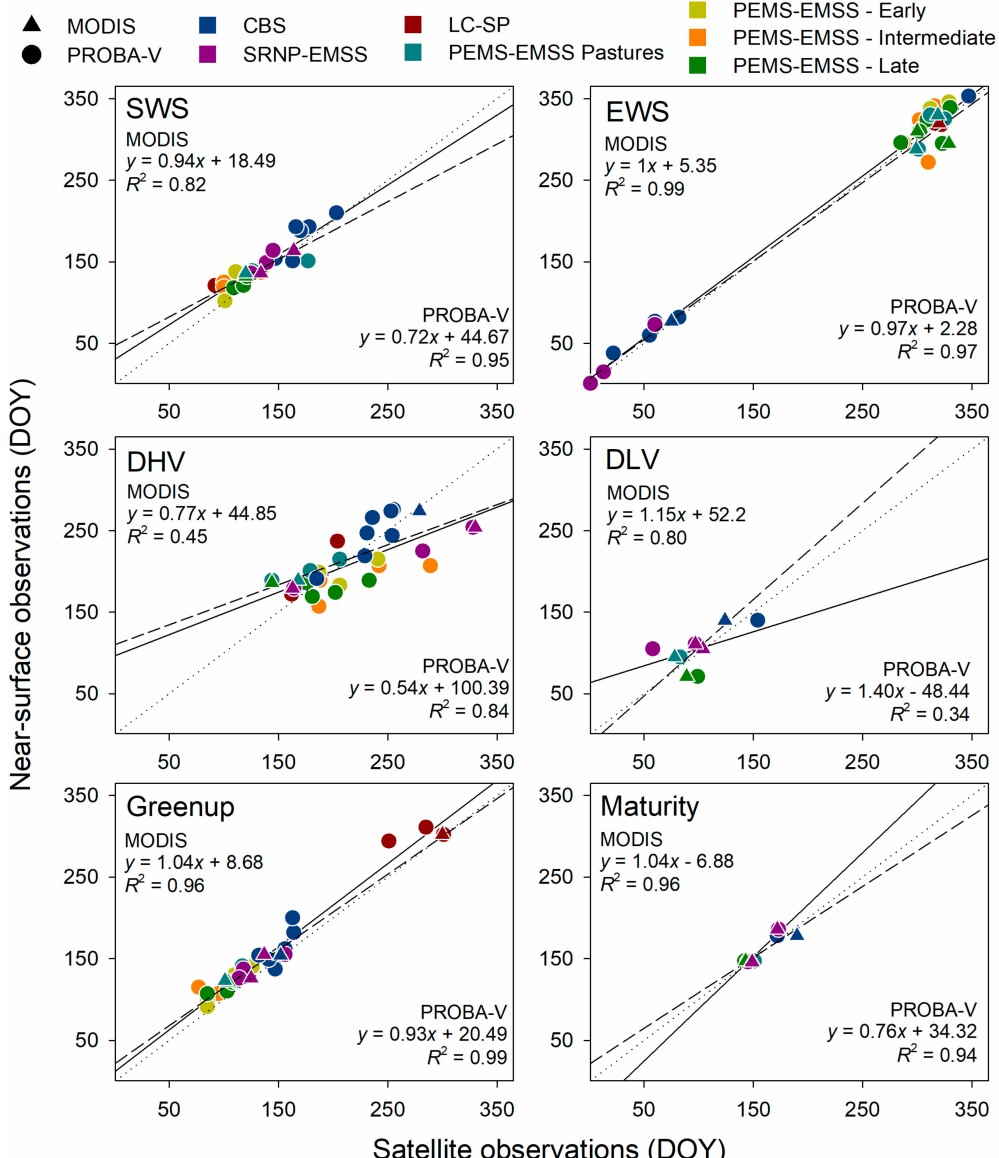

**Figure 5.** Scatterplot comparing the day of the year (DOY) of phenological events derived from satellite and near-surface observations at four tropical dry forests. The linear relationship between MODIS and near-surface observations is described by the solid lines, while the relationship between PROBA-V and near-surface observations is described by the dashed lines. Start and end of the wet season (SWS, EWS); day of the highest and lowest NDVI values (DHV, DLV). Acronyms for the sites are described on Table 1.

**Table 4.** Descriptors from the test comparing slopes from 1 of the relationship of phenometrics derived from satellite (MODIS and PROBA-V) and near-surface observations. In all models, each satellite sensor was analyzed as a predicted variable. The value *df* represents the degrees of freedom, SWS and EWS represent the start and the end of the wet season, and DHV and DLV represent the day of the highest and lowest values of NDVI.

| Phenometrics | MODIS | | | PROBA-V | | |
|---|---|---|---|---|---|---|
| | r-Statistics | *df* | *p*-Value | r-Statistics | *Df* | *p*-Value |
| SWS | −0.14 | 25 | 0.49 | −0.83 | 3 | 0.09 |
| EWS | 0.01 | 24 | 0.99 | −0.13 | 4 | 0.80 |
| DHV | −0.33 | 26 | 0.08 | −0.86 | 3 | 0.06 |
| DLV | 0.29 | 27 | 0.13 | 0.39 | 3 | 0.51 |
| Green-up | 0.19 | 25 | 0.33 | −0.51 | 4 | 0.30 |
| Maturity | 0.20 | 24 | 0.33 | −0.75 | 3 | 0.15 |

## 4. Discussion

Overall, our results reveal the variability existing between satellite-based NDVI observations when compared with high temporal resolution in situ measurements taken at optical phenology towers. This variability is mainly described by the temporal dissimilarities of NDVI$_S$ between near-surface and satellite observations, daily differences in the NDVI$_S$ trends, and the lack of correspondence of some phenometrics derived from near-surface and satellite sensors. However, as our results suggest, the above depends on the satellite sensor, the site of observation, and on meteorological and ecological conditions around the observations.

### 4.1. NDVI$_S$ Dissimilarities between Phenological Towers and Satellite Observations

Despite that the differences in magnitude of NDVI between sensors are compensated through their standardization, the comparisons of NDVI$_S$ using DTW reveal that time series derived from satellite observations tend to differ from those derived from near-surface sensors. This dissimilarity between NDVI$_S$ sequences could be the result of small variations in the phenological trends that can be observed by near-surface sensors, but not by the satellite products. These small variations may not be tracked by the satellite products due to their temporal resolution as well as the data preprocessing applied to the time series (e.g., smoothing). Likewise, the comparisons of NDVI$_S$ sequences using DTW also reveal that MODIS tends to present higher dissimilarities (37.5 ± 16.8) than PROBA-V (17.8 ± 14.5). Although our aim was not to make comparisons between sensors, this lack of similarity from MODIS is unexpected since the combination of both Terra and Aqua produce a slightly higher temporal resolution product (8 days) compared to PROBA-V (10 days). The lower similarity of the NDVI$_S$ from MODIS-near-surface comparisons could be attributed to the length of their time series. That is, temporally broader time series are more likely to differ compared to shorter ones. In general, the impact of time series length on DTW has been discussed by Folgado et al. [55]. Therefore, in order to describe which satellite sensors present the lower dissimilarity with near-surface observations, time series with similar lengths and temporal resolutions should be used.

### 4.2. Differences between Phenological Towers and Satellite Observations of Standardized NDVI

In general, differences in the NDVI magnitudes, observed in Figure 2, reinforce the idea of conduct standardization of the NDVI before comparing trends between satellite and near-surface observations. On these comparisons, although the overall distributions of NDVI$_{DS}$ are close to 0, our results show some that these distributions tend to be affected by season and the ENSO. Specifically, the seasonal effect of vegetation indices has been associated with variations in illumination geometry [16,56,57], where indices such as the enhanced vegetation index tend to be more affected than NDVI [57]. In general, near-infrared light tends to be more scattered when encountering vegetation at higher solar zenith angles than red light [46], causing increases in the NDVI depending on the season [57]. Since the near-surface observations sensors are fixed and their records were filtered to avoid possible high solar

zenith angles, the seasonal anisotropy of the satellite measurements may explain the differences in $NDVI_{DS}$ between the dry and wet season. This also may explain why the SRNP-EMSS, our observation site closest to the equator, did not present the seasonal effect on $NDVI_{DS}$ as observed in the other six sites. Despite this, anisotropy effects do not explain the lack of consistency of the distribution of higher or lower $NDVI_{DS}$ values in a given season or ENSO year. This is particularly important for the study of anomalies looking at the differences of observations in a given time and location in comparison with their climatology [48]. For example, it has been described that hydroclimatic events such as seasons or droughts oscillations are strongly correlated with NDVI anomalies [50]. However, it has not been described, to the authors' knowledge, how comparable are the NDVI anomalies derived from satellite and near-surface observations. The $NDVI_{DS}$ variability in a given ENSO year could be attributed to changes to in situ vegetation density, atmospheric or anisotropy effects on the satellite observations, and/or the distribution of samples (days of observations) in a given climatic event.

### 4.3. Comparisons of Phenometrics Derived from Satellite and Near-Surface Observations

The comparisons reveal that the DOY of some phenometrics derived from MODIS and PROBA-V tend to have a strong relationship with those derived from near-surface observations. Specifically, the stronger relationships are found on those phenometrics calculated from the NDVI derivatives (e.g., SWS, EWS, Green-up, Maturity). In general, these results are congruent with the existing body of literature [6,7,18,47,58], which partially suggests the phenometrics derived from NDVI satellite products are strongly correlated with those from ground observations. However, our results also reveal that phenometrics derived from satellite observations associated with the DOY with the highest and lowest NDVI values tend to be poorly associated with those derived from tower observations; probably due to the lower temporal resolution of these satellite products and their preprocessing, as explained before. These findings may suggest that if NDVI is considered as a proxy for primary productivity [59], values from satellite sensors could be poor descriptors of the days of maximum productivity or higher dormancy. Likewise, these results may also suggest that the utility of using NDVI time series to study drought anomalies by temporal peaks and valleys of NDVI could be compromised.

### 4.4. Recommendations and Future Perspectives for the Comparisons of NDVI between Satellite and Near-Surface Sensors

The inconsistencies observed between satellite and near-surface sensors could be the product of changes on the magnitude of the NDVI derived from satellite observations. The former is based on the poor correspondence of DHV and DLV between sensors and as a result of $NDVI_S$ differences between sensors during different meteorological events that take place at our study sites. Due to the anisotropy effects of the NDVI discussed in previous sections, and the natural variations of climatic events, we recommend considering the temporal inconsistencies of NDVI satellite-based observations to address the impact of drought events on TDF phenology. Future studies should also address how consistent NDVI anomalies, derived from satellite sensors, are when compared with near-surface observations. In addition, future research should consider direct comparisons using daily satellite products in order to detect phenological changes that occur during short time periods. Moreover, future inter-sensor comparisons must factor for illumination conditions and observation angles to avoid NDVI anisotropy.

## 5. Conclusions

Our results reveal differences in the temporal variations of satellite-based NDVI products from MODIS and PROBA-V in comparison with those from high temporal resolution optical phenology towers. These differences appear through the temporal discrepancies of the NDVI trends observed with the DTW analysis, as well as $NDVI_{DS}$ that are affected by seasons and ENSO years. These differences reinforce the idea of the susceptibility of NDVI satellite products to view-illumination geometries [57], which are critical in sites under strong seasonal variation and at higher latitudes. However, phenometrics

derived from NDVI satellite products tend to agree with those derived from optical phenology towers when these are associated with dates of shifts of the NDVI trends. Overall, the information derived from satellite-based NDVI products should be interpreted with great caution and implemented in ecosystems with a well-established knowledge of the vegetation phenology. The large-scale use of satellite-based NDVI products as an input for studies associated with productivity and anomalies should consider the anisotropy effects of this vegetation index.

**Supplementary Materials:** The following are available online at http://www.mdpi.com/2072-4292/11/19/2316/s1, Figure S1: Unprocessed and processed time series of Normalized Difference Vegetation Index (NDVI) derived from MODIS (Aqua and Terra) at four Tropical Dry Forests distributed along the Americas. Figure S2: Unprocessed and processed time series of Normalized Difference Vegetation Index (NDVI) derived from PROBA-V at four Tropical Dry Forests distributed along the Americas.

**Author Contributions:** Conceptualization, J.A.G. and G.A.S.-A.; Data Curation, methodology, formal analysis, and investigation, J.A.G.; Resources, G.A.S.-A.; Writing—Original Draft Preparation, J.A.G.; Writing—Review & Editing, G.A.S.-A., M.M.E.-S., and J.A.G.; Supervision, G.A.S.-A.; Funding acquisition, G.A.S.-A. and M.M.E.-S. All the authors agree with the final version of the manuscript.

**Funding:** This work was supported by the Natural Science and Engineering Research Council of Canada (NSERC) Discovery grant program, the Inter-American Institute for Global Change Research (IAI) CRN3-025 Tropi-Dry, and the Brazilian agencies FAPEMIG and CNPq who helped with the support of the maintenance of the towers. JAG is a Vanier Scholar supported by NSERC.

**Acknowledgments:** We thank the Saulo Castro, Cassidy Rankine, Eugenia Gonzales, Mike Hesketh and several Costarrican and Brazilian field technicians that help with data collected over the years. Thanks to the four anonymous reviewers for the useful suggestions and observations.

**Conflicts of Interest:** The authors declare no conflict of interest.

**Data Availability:** Preprocessed data of daily NDVI observations from the phenological towers are available at the Tropi-Dry dataverse through https://doi.org/10.7910/DVN/BDCJNP.

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
