# Peer review of "MODIS and PROBA-V NDVI Products Differ when Compared with Observations from Phenological Towers at Four Tropical Dry Forests in the Americas"

_remotesensing, doi:10.3390/rs11192316_

Round 1
Reviewer 1 Report
Overall this paper is very well organized and well written. I was interested to read this as ground-based validation of satellite retrievals for this specific land cover class appears to be lacking attention in the literature.
That said, I have major concerns about the conclusions drawn in this paper. I think they are at best misleading, and at worse simply incorrect.
Major issues:
1) The entire analysis on "temporal distortions" is very difficult to interpret. In the abstract it suggests that the satellite-based temporal series could be between 1 and 3 months off. This is very misleading to interpret, as the time series comparisons shown in Figure 2 appear to mirror fairly well and additionally the phenological DOY markers compare very well too. I fear that readers will not understand this metric and it overstates a minor discrepancy.
2) Line 31 of the abstract ("inadequate descriptor") is way too strong of a conclusion based on the results presented here.
3) Section 2.3: Were the actual dates of MODIS data acquistion used, or just the 8 day cumulative?
4) Section 2.5: More discussion/explanation for downscaling to 1 day temporal resolution is needed, perhaps include a figure showing the original data along with the downscaled data so the reader can understand the uncertainty imposed by this downscaling.
5) Section 2.6: Assumption (ii) is very strong. The tower spatial footprint is 17600 m^2, vs. 62500 m^2 for MODIS and 110889 m^2 for PROBA-V. This is a 3-6 fold different in footprints.
6) Section 3.1: Was this analysis done with the normalized values? It is not clear. As mentioned above, this metric is very difficult to interpret. Furthermore, it is not at all fair to compare MODIS and PROBA-V because the domain of temporal overlap with tower observations is very different. This is in fact mentioned on Lines 347-352, but still conclusions are being stated that are unfounded based on the results presented here.
7) Section 3.2 needs revision. There are conflicting statements (e.g. Lines 270-272 vs. Lines 281-282). Further the results of LC-SP and PEMS-EMSS Late conflict with the statement in Line 271. The sentence on Lines 273-276 is also not consistent with Figures and Tables. It is difficult to assess how the authors were comparing distributions, was it from the means? If so put the means in a table that is easier for the reader to compare.
8) The entire conclusion about poor matchups with DHV and DLV appears unfounded. Firstly, the matchups with DLV are not that terrible. Secondly, the matchups of DHV appear to be done incorrectly. For instance, the grouping in the upper left panel of Figure 5c indicates to me that the satellite was giving max values near the end of the previous calendar year, which in fact is not so far from the next year's early DOYs. DOY is not fair for seasons that cross December/January. Consider instead starting the count from SWS instead of default to Jan 1.
9)Lines 363-365: One station is not sufficient to make this conclusion.
10) Lines 372-375: I do not see the evidence to support this shown. Yes, the magnitude of NDVI is different but there is still anomaly to examine.
11) Lines 383-392: Disagree with all these, the results are not strong enough to entirely discount the utility of satellite observations.
Minor issues:
1) Edit title to remove 's' from "towers" and add 's' to "Forest"
2) Line 82: Reword. Is your hypothesis really that the products will fail? Makes you seem biased.
3) Add column to Table 1 for tower height.
4) Add more discussion in Section 2.6.2 about the need for normalization, it is interesting that such magnitude differences required this adjustment.
5) Section 2.6.3: By "local", does that mean "annual"? Also, please comment on the accuracy of the Tan method, as so many alternative methods exist.
6) Lines 339-341: This is in fact the opposite of what one would expect. One would expect that finer temporal resolution would yield closer agreement to ground observations.
Author Response
Reviewer 1
Overall this paper is very well organized and well written. I was interested to read this as ground-based validation of satellite retrievals for this specific land cover class appears to be lacking attention in the literature.
That said, I have major concerns about the conclusions drawn in this paper. I think they are at best misleading, and at worse simply incorrect.
Major issues:
1) The entire analysis on "temporal distortions" is very difficult to interpret. In the abstract it suggests that the satellite-based temporal series could be between 1 and 3 months off. This is very misleading to interpret, as the time series comparisons shown in Figure 2 appear to mirror fairly well and additionally the phenological DOY markers compare very well too. I fear that readers will not understand this metric and it overstates a minor discrepancy.
R/. Changes applied. We rewrote this section as well as their methods and the interpretation of the results in the discussion.
2) Line 31 of the abstract ("inadequate descriptor") is way too strong of a conclusion based on the results presented here.
R/. Changes applied. It was corrected to ‘inconsistent’.
3) Section 2.3: Were the actual dates of MODIS data acquistion used, or just the 8 day cumulative?
R/. As section 2.3 stated, we used a combination of both MOD13Q1 and MYD13Q1 satellite-products. These are level-3 products of 16-day time-series observations. When we combine then, it turns into an 8-day time series. These are not cumulative values. The dates used were the dates provided by Oak Ridge National Laboratory DAAC MODIS land process subset tool. No changes applied.
4) Section 2.5: More discussion/explanation for downscaling to 1 day temporal resolution is needed, perhaps include a figure showing the original data along with the downscaled data so the reader can understand the uncertainty imposed by this downscaling.
R/. Some changes applied. We consider that the explanation of why downscaling the satellite observations are already embedded on lines 161-162. However, we also consider that a figure showing the data (?) before and after the application of the downscaling may help the reader understand their importance. A figure showing the unprocessed and processed observations were added in supplementary materials.
5) Section 2.6: Assumption (ii) is very strong. The tower spatial footprint is 17600 m^2, vs. 62500 m^2 for MODIS and 110889 m^2 for PROBA-V. This is a 3-6 fold different in footprints.
R/. Yes, the reviewer is right, this assumption is very strong. Unfortunately, there is no other way to do it. It is unlikely and difficult to build a tower for a given height to cover a large pixel area, as well as it is very expensive to build several towers around a given area to cover a large pixel. The selection of sites to build the towers was based on the homogeneity of the forests; therefore, we can assume that the area covered by the near-surface sensor is representative of the area covered by the pixel. No changes applied.
6) Section 3.1: Was this analysis done with the normalized values? It is not clear. As mentioned above, this metric is very difficult to interpret. Furthermore, it is not at all fair to compare MODIS and PROBA-V because the domain of temporal overlap with tower observations is very different. This is in fact mentioned on Lines 347-352, but still conclusions are being stated that are unfounded based on the results presented here.
R/. In the first submission, we did it using non-normalized values; however, in this one, we change it to normalized NDVI. Currently, this is explicitly stated in section 2.6.1. The methods of normalization or “standardization”, as defined by us, are now described in section 2.5. Changes in the discussion were performed to avoid this confusion.
7) Section 3.2 needs revision. There are conflicting statements (e.g. Lines 270-272 vs. Lines 281-282). Further the results of LC-SP and PEMS-EMSS Late conflict with the statement in Line 271. The sentence on Lines 273-276 is also not consistent with Figures and Tables. It is difficult to assess how the authors were comparing distributions, was it from the means? If so put the means in a table that is easier for the reader to compare.
R/. Changes applied. These sentences were rewritten. The comparisons were performed on the overall distribution (higher or lower) if the test rejects the null hypothesis (no differences). We consider that this can not be done using mean values due to the temporal autocorrelation of the samples.
8) The entire conclusion about poor matchups with DHV and DLV appears unfounded. Firstly, the matchups with DLV are not that terrible. Secondly, the matchups of DHV appear to be done incorrectly. For instance, the grouping in the upper left panel of Figure 5c indicates to me that the satellite was giving max values near the end of the previous calendar year, which in fact is not so far from the next year's early DOYs. DOY is not fair for seasons that cross December/January. Consider instead starting the count from SWS instead of default to Jan 1.
R/. Changes applied. The values were reviewed, and we agree with the reviewer’s comment. In general, the starting day of the year was established as to July 1st for sites at the southern hemisphere. With this method, we corrected the effect of phenometrics close to the beginning or the end of the calendar year.
9) Lines 363-365: One station is not sufficient to make this conclusion.
R/. We agree that one station is not enough. However, this is not a conclusion; this is a statement to explain the sources of variability. We are using hedging verbs to soften our statement. No changes applied.
10) Lines 372-375: I do not see the evidence to support this shown. Yes, the magnitude of NDVI is different but there is still anomaly to examine.
R/. Some changes applied. Here our point is that traditional procedures to study anomalies using local temporal normalizations could be affected by the temporal inconsistencies of observations between sites. These inconsistencies are described in figure 3 and 4. Yes, the magnitude of NDVI is different and we may observe local temporal anomalies; however, are these anomalies consistent between sites during different seasons or ENSO months? Our results do not support that. We rewrote this sentence.
11) Lines 383-392: Disagree with all these, the results are not strong enough to entirely discount the utility of satellite observations.
R/. Some changes applied. We do not suggest to entirely discount the utility of satellite observations. We suggest that the application of these products should be interpreted with caution because there are temporal inconsistencies between the observations from the satellite and near-surface sensors. We rewrote some sentences in the discussion.
Minor issues:
1) Edit title to remove 's' from "towers" and add 's' to "Forest"
R/. Changes applied. The title was corrected.
2) Line 82: Reword. Is your hypothesis really that the products will fail? Makes you seem biased.
R/. That was not our intention. Changes were performed to this paragraph to avoid this confusion.
3) Add column to Table 1 for tower height.
R/. Changes applied. A column describing the canopy height was added to the table.
4) Add more discussion in Section 2.6.2 about the need for normalization, it is interesting that such magnitude differences required this adjustment.
R/. Changes applied. Now the explanation of the standardization of NDVI is explained in section 2.5. Likewise, we added references to guide the reader about how the NDVI could be affected by the sensors properties and the interaction of light with the surfaces.
5) Section 2.6.3: By "local", does that mean "annual"? Also, please comment on the accuracy of the Tan method, as so many alternative methods exist.
R/. Changes applied. Yes. We added ‘annual local maximum and minimum of the first derivative’ to make a reference to the lowest and highest values of the derivative each year.
6) Lines 339-341: This is in fact the opposite of what one would expect. One would expect that finer temporal resolution would yield closer agreement to ground observations.
R/. Correct, our result was unexpected. In fact, we stated in this section that “This lack of similarity from MODIS is unexpected since the combination of both Terra and Aqua produce a slightly higher temporal resolution product (8-days) in comparison with PROBA-V (10-days)”. No changes applied.
Reviewer 2 Report
The manuscript titled “MODIS and PROBA-V NDVI products differ when compared with phenological towers observations at four Tropical Dry Forest in the Americas” is well-written, well-organized, and I believe describes some valuable and insightful work that would be of interest to many in the field of time series remote sensing for vegetation. I have recommended it for publication pending satisfactory response to reviewer comments and suggestions. I do not have very major changes to recommend, but I do encourage the authors to expand on a few of their explanations, and deepen (or add) their discussions on some particular issues. Below are my specific comments.
--- Specific Comments ---
Title: i) it should be “with phenological tower observations” or “observations from phenological towers”, and ii) with this title do you mean to emphasize the difference between the sensors themselves (in their relationship to ground observations), or to emphasize the difference between satellite and ground observations in general? This title suggests the former to me. And if that is the case, then I would expect to see far more focus in your Discussion section on the differences in the results between the sensors.
Ln 14: should be either “at large scales” or “at a large scale”
Ln 18: should be “predict vegetation phenology”
Ln 25: I would suggest “or when the El Nino…”
Ln 28: use consistent terminology – stick to “phenometrics” rather than all of a sudden using “phenophases” (in the paper itself, if there is a different meaning then this can be explained/defined and the terms each used in appropriate places)
Ln 28: do you mean “lower” or “lowest”; see further comments below on this
Ln 41: are the Essential Biodiversity Variables an official term for a list of variables defined by a particular group/organization? If so, then please indicate in the text which group or organization (e.g., “are considered by xxx to be part of a suite of….”). If not, then there is no need to capitalize
Ln 53: should “than” be “that (tends to scatter [*when encountering vegetation* – adding this would be informative])”?
Ln 53: should be “relationship between NDVI and…”
Ln 55: for readers, here please expand/briefly describe why the nature of satellite NDVI is limited in its validation (coarse spatial resolution, broad spatial and temporal coverage… )
Ln 62: should be “the application *of vegetation *indices” – you use both indexes and indices in your paper; I believe ‘indices’ is more proper but stick with one and be consistent throughout
Ln 70: should be “is essential to unraveling key….”
Ln 71: I would suggest “such as the El Nino…”
Ln 73: I believe you mean “how well…. predict *ground-based NDVI variability….” This should be specified
Lns 86 & 88: should be “at four *TDFs in…” and “four different *TDF communities”
Ln 102: “between them”
Ln 103: you mean “5-6 meters above the forest canopy height”, right? Perhaps would be advisable to specify here
Ln 124: what were the over-pass times for the MODIS Terra and Aqua and PROBA-V satellites? Does it line up with this 10:00 to 14:00 time window? Please explain here. And if not, please address the implications of this for your results and conclusions, in the Discussion section. Also, how do the over-pass times for the different sites differ, or differ between sensors? This should also be described/listed in your methods and discussed in the Discussion section, particularly since you are comparing with ground observations.
Ln 145: I would suggest: “As are MODIS products, these…”
Ln 149: “From these….”
Ln 150: was there any concern for pixel location shifts between images for the particular pixels you chose? Are the flux towers located right in the center of each pixel? Or are some of them close to pixel edges? In the Discussion section, please address this issue and its possible implications for your results/conclusions.
Ln 185: I would suggest: “In general, the DTW analysis enables one to describe the quality of the global alignment between two time series.”
Ln 195: what “minimum” and “maximum” NDVI values do you use here? Are they maximums and minimums from the entire multi-annual time series for each sensor and pixel? Or something else? Please clarify.
Ln 223: Here I would like to see i) a definition of “phenometrics” (i.e., does it simply stand for “phenological metrics”?), and ii) the consistent use of this term in both the main paper and any captions (i.e., I would not recommend substituting it with “phenophases” here and there unless you explicitly state that these both refer to the same thing).
Ln 229: why is Greenup capitalized (and only sometimes), while the other phenometrics are not? Be consistent. Either capitalize all of them, as formal terms, or leave them all lower-case throughout.
Ln 234: sometimes you use “lowest NDVI” and sometimes you use “lower NDVI”. Do these refer to the same thing? Again, be consistent. Also, there is a difference between these two. Be sure to represent the metric itself with the appropriate term.
Ln 235: why did you average the phenometrics across years?? Wouldn’t it be more informative to compare on a year-by-year basis? Please explain your reasons
Ln 249: is there a unit for these distances?? Please include. I believe it is days, is it not?
Ln 266: should be “the density distributions of…”
Ln 272: I think you mean “An example of this trend is present…” The word “expectation” doesn’t really make sense here to me
Ln 300: should be “seasonal or ENSO effects”
Ln 302: should be “Observations… come from…”
Ln 304: please write out the meaning of the DOY acronym the first time it appears in the main text of the paper
Ln 310 – 312: the results to me do not seem to suggest that the DLV phenophase is nearly as poorly correlated/modeled as the DHV, either in Figure 5, or in Table 4 – the R2 and RSE values are much higher for DLV. Please reconsider your interpretation and discussion on this point in the paper
Figure 5: here “higher” and “lower” are used instead of “highest” and “lowest”. This is confusing.
Table 4: please use the same DHV and DLV acronyms here as used elsewhere in the paper
Discussion Section: I strongly recommend that much more attention be given to the differences between the two, and these addressed/discussed in more depth, particularly in subsections 4.2 and 4.3. The title suggests to me that this is a key component of the work.
For instance, the time series from the two sensors in Figure 2 do not seem to be so very different from one another visually, but from the DTW analysis they appear to be numerically. Can you address why this is?
Also, please explore any differences in over-pass times between the sensors and how this might play a role in your results.
Are the effects of season or ENSO events stronger in the NDVI time series from one sensor over the other? Discuss.
In terms of results for phenometrics, please explore differences in the results between the two sensors – e.g., particularly for DHV and Maturity
Based on your results and observations, do you have recommendations for future users with regard to the use of one sensor over the other, either in general or under particular conditions?
Ln 359-360: both “indexes” and “indices” are used in the same sentence. As already suggested, pick one and be consistent.
Ln 368 – 370: this sentence (“However, it has not…”) is confusing to me. Do you mean: “it has not been described… how NDVI anomalies derived from satellite observations compare to variations in ground-based NDVI observations.”? Please clarify.
Ln 399: awkwardly worded sentence. Perhaps: “These difference reinforce the susceptibility of NDVI satellite products to the effects of view-illumination geometries.”
Author Response
Reviewer 2
The manuscript titled “MODIS and PROBA-V NDVI products differ when compared with phenological towers observations at four Tropical Dry Forest in the Americas” is well-written, well-organized, and I believe describes some valuable and insightful work that would be of interest to many in the field of time series remote sensing for vegetation. I have recommended it for publication pending satisfactory response to reviewer comments and suggestions. I do not have very major changes to recommend, but I do encourage the authors to expand on a few of their explanations, and deepen (or add) their discussions on some particular issues. Below are my specific comments.
--- Specific Comments ---
Title: i) it should be “with phenological tower observations” or “observations from phenological towers”, and ii) with this title do you mean to emphasize the difference between the sensors themselves (in their relationship to ground observations), or to emphasize the difference between satellite and ground observations in general? This title suggests the former to me. And if that is the case, then I would expect to see far more focus in your Discussion section on the differences in the results between the sensors.
R/. Changes applied. ‘Observations from phenological towers’ is used. Sections of the discussion were also modified.
Ln 14: should be either “at large scales” or “at a large scale”.
R/. Changes applied. It was modified to ‘at large scales’.
Ln 18: should be “predict vegetation phenology”
R/. Changes applied. It was modified to ‘predict vegetation phenology’.
Ln 25: I would suggest “or when the El Nino…”
R/. Here we want emphasize both El Niño and La Niña. No changes applied.
Ln 28: use consistent terminology – stick to “phenometrics” rather than all of a sudden using “phenophases” (in the paper itself, if there is a different meaning then this can be explained/defined and the terms each used in appropriate places)
R/. Changes applied. The term ‘phenometrics’ is consistently used throughout the manuscript.
Ln 28: do you mean “lower” or “lowest”; see further comments below on this
R/. Changes applied. We meant ‘lowest’.
Ln 41: are the Essential Biodiversity Variables an official term for a list of variables defined by a particular group/organization? If so, then please indicate in the text which group or organization (e.g., “are considered by xxx to be part of a suite of….”). If not, then there is no need to capitalize
R/. Changes applied. Thanks for the suggestion. The sentence was modified as recommended.
Ln 53: should “than” be “that (tends to scatter [*when encountering vegetation* – adding this would be informative])”?
R/. Changes applied.
Ln 53: should be “relationship between NDVI and…”
R/. No changes applied. We consider that ‘with’ is the proper preposition instead of the conjunction ‘and’.
Ln 55: for readers, here please expand/briefly describe why the nature of satellite NDVI is limited in its validation (coarse spatial resolution, broad spatial and temporal coverage… )
R/. Changes applied. We added a sentence and a reference to expand this idea.
Ln 62: should be “the application *of vegetation *indices” – you use both indexes and indices in your paper; I believe ‘indices’ is more proper but stick with one and be consistent throughout
R/. Changes applied. Thanks for your comment, we used ‘indices’ through the manuscript.
Ln 70: should be “is essential to unraveling key….”
R/. Changes applied.
Ln 71: I would suggest “such as the El Nino…”
R/. Again we consider that ENSO involves both El Niño and La Niña. No changes applied.
Ln 73: I believe you mean “how well…. predict *ground-based NDVI variability….” This should be specified
R/. Changes applied. This sentence was modified.
Lns 86 & 88: should be “at four *TDFs in…” and “four different *TDF communities”
R/. Changes applied.
Ln 102: “between them”
R/. Changes applied.
Ln 103: you mean “5-6 meters above the forest canopy height”, right? Perhaps would be advisable to specify here
R/. Changes applied. Thanks for the suggestion.
Ln 124: what were the over-pass times for the MODIS Terra and Aqua and PROBA-V satellites? Does it line up with this 10:00 to 14:00 time window? Please explain here. And if not, please address the implications of this for your results and conclusions, in the Discussion section. Also, how do the over-pass times for the different sites differ, or differ between sensors? This should also be described/listed in your methods and discussed in the Discussion section, particularly since you are comparing with ground observations.
R/. Some changes applied. The over-pass time for MODIS and PROBA-V satellites may or may not line up with this time window. Due to the overlap of the satellites with polar orbits, multiple observations may exist from the same day and same pixel, and more land tends to be covered as the satellite get close to the equator. Specifically, for the case of MOD13Q1 and MYD13Q1, these products used pre-composite data observations and applied a series of filters based on quality, cloud presence, and viewing geometry. It retains in a fixed grid the best quality pixel, which is a cloud-free, nadir view pixel with no residual atmospheric contamination. The aim of this is to extract a single value per pixel from all the retained filtered data; which will represent at the end the pixel for the particular 16-day period. A similar principle applies in the PROBA-V NDVI product. We added in the methods section more explanation about these products. Likewise, in the discussion section we added sentences of how the temporal resolution and preprocessing applied to these products may not detect small changes in the forest phenology.
Ln 145: I would suggest: “As are MODIS products, these…”
R/. Changes applied. This sentence was corrected to ‘Similar to MODIS products, …’.
Ln 149: “From these….”
R/. Changes applied.
Ln 150: was there any concern for pixel location shifts between images for the particular pixels you chose? Are the flux towers located right in the center of each pixel? Or are some of them close to pixel edges? In the Discussion section, please address this issue and its possible implications for your results/conclusions.
R/. According to the products guide, there is a lack in the location shift of pixels in these products since these are created using a fixed grid for the extraction of ‘best pixels’ in a compilation of observations. However, most of our towers tend to be located in the center of the pixels and an example of this can be observed in figure 1 of Rankine et al. 2017 for the PEMS-ESSS towers. Since there are no temporal shifts in the location of pixels, we consider that this may not have important or possible implications in our results. No changes applied.
Rankine, C., Sánchez-Azofeifa, G. A., Guzmán, J. A., Espirito-Santo, M. M., & Sharp, I. (2017). Comparing MODIS and near-surface vegetation indexes for monitoring tropical dry forest phenology along a successional gradient using optical phenology towers. Environmental Research Letters, 12(10), 105007.
Ln 185: I would suggest: “In general, the DTW analysis enables one to describe the quality of the global alignment between two time series.”
R/. Changes applied. The sentence was corrected as the reviewer suggested.
Ln 195: what “minimum” and “maximum” NDVI values do you use here? Are they maximums and minimums from the entire multi-annual time series for each sensor and pixel? Or something else? Please clarify.
R/. Changes applied. We added extra sentences describing the application of the formula.
Ln 223: Here I would like to see i) a definition of “phenometrics” (i.e., does it simply stand for “phenological metrics”?), and ii) the consistent use of this term in both the main paper and any captions (i.e., I would not recommend substituting it with “phenophases” here and there unless you explicitly state that these both refer to the same thing).
R/. Some changes applied. The term ‘phenometrics’ is consistently used throughout the manuscript. It stands for phenological metrics. We did not provide a definition since these are metrics to describe a phenomenon.
Ln 229: why is Greenup capitalized (and only sometimes), while the other phenometrics are not? Be consistent. Either capitalize all of them, as formal terms, or leave them all lower-case throughout.
R/. Changes applied. Lower-case was used throughout the text.
Ln 234: sometimes you use “lowest NDVI” and sometimes you use “lower NDVI”. Do these refer to the same thing? Again, be consistent. Also, there is a difference between these two. Be sure to represent the metric itself with the appropriate term.
R/. Changes applied. Consistencies were reviewed through the manuscript.
Ln 235: why did you average the phenometrics across years?? Wouldn’t it be more informative to compare on a year-by-year basis? Please explain your reasons
R/. Changes applied. In the first submission, we performed an averaging per site with the aim to avoid the effect of the number of observations per site in the regressions. That is, sites with more years of observation will have more weight in the regression results. However, we understand the reviewer concern, and in this new version we performed regressions using all observations. Likewise, we performed a standardize major axis regressions to get an accurate estimation of the intercept and slope. Within this analysis, we also compare the slope form 1 (perfect prediction) in order to know if the estimated slope differs from a perfect scenario 1:1.
Ln 249: is there a unit for these distances?? Please include. I believe it is days, is it not?
R/. No, the DTW distance has no units since it comes from comparisons between dimensionless sequences. No changes applied.
Ln 266: should be “the density distributions of…”
R/. Changes applied.
Ln 272: I think you mean “An example of this trend is present…” The word “expectation” doesn’t really make sense here to me
R/. Changes applied. This sentence was modified.
Ln 300: should be “seasonal or ENSO effects”
R/. Changes applied.
Ln 302: should be “Observations… come from…”
R/. Changes applied.
Ln 304: please write out the meaning of the DOY acronym the first time it appears in the main text of the paper
R/. Changes applied.
Ln 310 – 312: the results to me do not seem to suggest that the DLV phenophase is nearly as poorly correlated/modeled as the DHV, either in Figure 5, or in Table 4 – the R2 and RSE values are much higher for DLV. Please reconsider your interpretation and discussion on this point in the paper
R/. Changes applied. Modifications to the analysis, figures and the interpretation of these metrics were performed.
Figure 5: here “higher” and “lower” are used instead of “highest” and “lowest”. This is confusing.
R/. Changes applied. A new figure was added.
Table 4: please use the same DHV and DLV acronyms here as used elsewhere in the paper
R/. Changes applied. The acronyms were corrected.
Discussion Section: I strongly recommend that much more attention be given to the differences between the two, and these addressed/discussed in more depth, particularly in subsections 4.2 and 4.3. The title suggests to me that this is a key component of the work.
R/. Changes applied. We corrected the title as our intention is not to compare satellite products because they present different features (e.g. location of bands, methods for the creation of the NDVI products, temporal resolution). In this new version we emphasized the differences between satellite and near-surface observations.
For instance, the time series from the two sensors in Figure 2 do not seem to be so very different from one another visually, but from the DTW analysis they appear to be numerically. Can you address why this is?
R/. Changes applied. Adjustments to the DTW analyses were performed as well as their interpretation in the discussion.
Also, please explore any differences in over-pass times between the sensors and how this might play a role in your results.
R/. Changes applied. In this new version, we added sentences in the discussion explaining how the temporal resolution and the preprocessing of the data may affect our results.
Are the effects of season or ENSO events stronger in the NDVI time series from one sensor over the other? Discuss.
R/. Our intention was not to compare satellite sensors because they present different features. No changes applied.
In terms of results for phenometrics, please explore differences in the results between the two sensors – e.g., particularly for DHV and Maturity
R/. Our intention was not to compare satellite sensors because they present different features. No changes applied. No changes applied.
Based on your results and observations, do you have recommendations for future users with regard to the use of one sensor over the other, either in general or under particular conditions?
R/. Again, Our intention was not to compare satellite sensors because they present different features. No changes applied. No changes applied.
Ln 359-360: both “indexes” and “indices” are used in the same sentence. As already suggested, pick one and be consistent.
R/. Changes applied. We used ‘indices’ throughout the text.
Ln 368 – 370: this sentence (“However, it has not…”) is confusing to me. Do you mean: “it has not been described… how NDVI anomalies derived from satellite observations compare to variations in ground-based NDVI observations.”? Please clarify.
R/. Changes applied.
Ln 399: awkwardly worded sentence. Perhaps: “These difference reinforce the susceptibility of NDVI satellite products to the effects of view-illumination geometries.”
R/. Changes applied. The sentence was partially modified as the reviewer suggested.
Reviewer 3 Report
This manuscript is focused on comparing MODIS and PROBA-V NDVI to near ground NDVI measurements at four Tropical Dry Forests. It is an interesting study to conduct and I believe it could be an interesting addon to the area. However, my main concern is that since the focus of the manuscript is to compare MODIS and PROBA-V NDVI to the daily ground observations, it makes much more sense to use daily/subdaily MODIS reflectances to calculate daily NDVI instead of using 16-day MODIS NDVI products. I would suggest the authors consider two kinds of MODIS reflectance products and I believe using either of them will be more convincing than using MOD13Q1 and MYD13Q1.
- Reflectance product 1:
MCD19A1, MODIS/Terra+Aqua Land Surface BRF Daily L2G Global 500m and 1km SIN Grid (https://ladsweb.modaps.eosdis.nasa.gov/missions-and-measurements/products/MCD19A1/ ), is the new reflectance product with advanced atmospheric correction and cloud detection. This product is the subdaily scale. If it is combined with
MCD19A3, MODIS/Terra+Aqua BRDF Model Parameters 8-Day L3 Global 1km SIN Grid ( https://ladsweb.modaps.eosdis.nasa.gov/missions-and-measurements/products/MCD19A3/ ), the BRDF-adjusted reflectances can be calculated, which could greatly reduce the anisotropy effect.
- Reflectance product 2:
MCD43A4, Nadir BRDF-Adjusted Reflectance (NBAR) Product (https://lpdaac.usgs.gov/products/mcd43a4v006/), also provides BRDF-adjusted reflectances at the daily scale.
Detailed comments and suggestions are shown below.
- L. 90-91: The sentence “The canopy height of the phenological and flux tower is close to 10.5 m.” is a little bit confusing. Since at the end of the paragraph, the authors mentioned that “The height of these towers varies between then; however, all these maintain a distance close to 5-6 meters from the canopy height.”, probably here the authors only need to describe canopy height.
- L. 105 Figure 1: Please provide the acronym of the sites in the figure to make the readers easy to find the sites mentioned in the article.
- L. 109: Table 1: Please also provide latitude, longitude, altitude, and canopy height information for each site. Dominant species and site references would be a plus.
- L. 129-142: section 2.3: please see the above main concern.
- L. 161: In this manuscript, the phenology tower observations serve as ground truth, so it should not be smoothed. And even the gap-filled points should not be used for the direct NDVI comparison.
- L. 162: Again, the daily/subdaily MODIS data need to be used and smoothing is not needed. Gap-filled points should not be used for the direct NDVI comparison as well.
- L. 166: The time length covering the moving window of 25 observations will vary according to the missing points, which probably will influence the smoothing.
- L. 184: Please provide more details on this DTW analysis, give out the equation, and the expected outputs and their meanings.
- L. 225: Add ‘[45]’ after ‘Tan et al. (2011)’
- L. 235-237: There is an issue here. It is not necessary to do the averages per site. Actually, after averaging, the linear regressions with only 5 or 7 points cannot give us a statistically significant outcome when the true effect is not zero. The authors should use the phenometrics for different years directly.
- L. 256: Figure 2: It probably would be easy for the readers to see more details in time series when having only one column instead of the two.
- L. 260: Table 2: As suggested previously, please try to give more details on the DTW algorithm and include the equations to help the readers to understand how the minimum global distance is calculated.
- L. 280: ‘Fig 3g’ should be ‘Fig 4g’.
- L. 284: Figure 3: Please include the statistical significance for column 1 as well.
- L. 292: Figure 4: Please include the statistical significance for column 1 as well.
- L. 299: Table 3: Please include F and p-value for column 1 in both figures 3 and 4.
- L. 304: This whole section needs rework and the last sub-figure in Figure 5 isn’t right based on the intercepts in Table 4.
- Discussion and Conclusions need to be revised accordingly based on the editing of Materials and methods and Results.
Author Response
Reviewer 3
This manuscript is focused on comparing MODIS and PROBA-V NDVI to near ground NDVI measurements at four Tropical Dry Forests. It is an interesting study to conduct and I believe it could be an interesting addon to the area. However, my main concern is that since the focus of the manuscript is to compare MODIS and PROBA-V NDVI to the daily ground observations, it makes much more sense to use daily/subdaily MODIS reflectances to calculate daily NDVI instead of using 16-day MODIS NDVI products. I would suggest the authors consider two kinds of MODIS reflectance products and I believe using either of them will be more convincing than using MOD13Q1 and MYD13Q1.
- Reflectance product 1:
MCD19A1, MODIS/Terra+Aqua Land Surface BRF Daily L2G Global 500m and 1km SIN Grid (https://ladsweb.modaps.eosdis.nasa.gov/missions-and-measurements/products/MCD19A1/ ), is the new reflectance product with advanced atmospheric correction and cloud detection. This product is the subdaily scale. If it is combined with MCD19A3, MODIS/Terra+Aqua BRDF Model Parameters 8-Day L3 Global 1km SIN Grid (https://ladsweb.modaps.eosdis.nasa.gov/missions-and-measurements/products/MCD19A3/ ), the BRDF-adjusted reflectances can be calculated, which could greatly reduce the anisotropy effect.
- Reflectance product 2:
MCD43A4, Nadir BRDF-Adjusted Reflectance (NBAR) Product (https://lpdaac.usgs.gov/products/mcd43a4v006/), also provides BRDF-adjusted reflectances at the daily scale.
R/. We understand the reviewer’s concerns; however, we have reasons to chose MOD13Q1 and MYD13Q1. As we state in the manuscript, ‘these products were selected over the daily-raw MODIS surface reflectance (Level 1) because they are more readily accessed and commonly used by the scientific community that works with vegetation phenology and temporal anomalies. Likewise, the spatial resolution of the products used in our study is half size than the recommended by the reviewer. Despite it is true that homogeneous forest covers were selected to build the towers, at large areas, factors such as the fragmentation or gaps may lead to mixing of different covers that may not represent that true vegetation phenology or successional stage (for the case of PNMS-EMSS). Likewise, the daily products mentioned by the reviewer (MCD19A1 and MCD43A4) are highly susceptible to the cloud cover, even though they are atmospheric corrected. This may lead to a high proportion of missing observations during the wet season that will need to be gap-filled and smoothed. In addition, the reviewer comment only applies to the MODIS observations, as near-surface comparisons with PROBA-V will also be affected by the temporal resolution.
In our manuscript, we acknowledged that different factors need to be contemplated in order to use near-surface observations as a strict ground-truth validator. Factors such as bandwidth used between sensors, location of bands, hours of data acquisition, incident-reflected angles, and spatial distribution of the observations may affect the NDVI observations. That is why we avoid the use of ‘validation’ throughout the manuscript. Our goal was to look to sources of variability and what kind of aspects we need to consider in order to study the phenology of the Tropical Dry Forests.
Detailed comments and suggestions are shown below.
90-91: The sentence “The canopy height of the phenological and flux tower is close to 10.5 m.” is a little bit confusing. Since at the end of the paragraph, the authors mentioned that “The height of these towers varies between then; however, all these maintain a distance close to 5-6 meters from the canopy height.”, probably here the authors only need to describe canopy height.
R/. Changes applied. This paragraph was modified.
- L. 105 Figure 1: Please provide the acronym of the sites in the figure to make the readers easy to find the sites mentioned in the article.
R/. Changes applied. This figure was modified.
- L. 109: Table 1: Please also provide latitude, longitude, altitude, and canopy height information for each site. Dominant species and site references would be a plus.
R/. Some changes applied. Latitude and longitude were not included since these are already represented in the map. However, tower height and altitude were included.
- L. 129-142: section 2.3: please see the above main concern.
R/. This is responded above.
- L. 161: In this manuscript, the phenology tower observations serve as ground truth, so it should not be smoothed. And even the gap-filled points should not be used for the direct NDVI comparison.
R/. Here we are using near-surface observations as a point of reference to determine the variability of satellite observations, not to validate them. We intentionally avoid the use of the term ‘validation’ throughout the manuscript because this implies a direct and fair comparison between sensors; that is: corrections of bandwidth, location of bands, hours of data acquisition, angles, and spatial distribution of the observations. Our time series can be smoothed because there are days with cloud cover, and these days affect the variability of NDVI from ground observations.
- L. 162: Again, the daily/subdaily MODIS data need to be used and smoothing is not needed. Gap-filled points should not be used for the direct NDVI comparison as well.
R/. Same response as above. No changes applied.
- L. 166: The time length covering the moving window of 25 observations will vary according to the missing points, which probably will influence the smoothing.
R/. Correct, that is why we first applied gap-filled and then smoothing. As we mentioned, these steps were performed to correct sporadic sensor failures from the phenological towers. No changes applied.
- L. 184: Please provide more details on this DTW analysis, give out the equation, and the expected outputs and their meanings.
R/. Changes applied. We rewrote this section as well as their interpretation in the discussion.
- L. 225: Add ‘[45]’ after ‘Tan et al. (2011)’
R/. Changes applied.
- L. 235-237: There is an issue here. It is not necessary to do the averages per site. Actually, after averaging, the linear regressions with only 5 or 7 points cannot give us a statistically significant outcome when the true effect is not zero. The authors should use the phenometrics for different years directly.
R/. Changes applied. In the first version of the manuscript, we used averages per site aiming to avoid the effect of the number of observations per site in the regressions. That is, sites with more years of observation would have more weight in the regression results. However, in this new version we performed regressions using all observations. Likewise, we performed standardized major axis regressions to get an accurate estimation of the intercept and slope. Within this analysis, we also compare the slope form 1 (perfect prediction) in order to know if the estimated slope differs from a perfect scenario 1:1.
- L. 256: Figure 2: It probably would be easy for the readers to see more details in time series when having only one column instead of the two.
R/. The figures are in high definition. Two columns were used in order to make efficient use of space. No changes applied.
- L. 260: Table 2: As suggested previously, please try to give more details on the DTW algorithm and include the equations to help the readers to understand how the minimum global distance is calculated.
R/. Changes applied as mentioned above.
- L. 280: ‘Fig 3g’ should be ‘Fig 4g’.
R/. Changes applied. Thanks for the observation.
- L. 284: Figure 3: Please include the statistical significance for column 1 as well.
R/. The statistical parameters of figure 3 are described in Table 3 as mentioned in the figure legend. There is no need to add more text to a figure. No changes applied.
- L. 292: Figure 4: Please include the statistical significance for column 1 as well.
R/. The statistical parameters of figure 3 are described in Table 3 as mentioned in the figure legend. There is no need to add more text to a figure. No changes applied.
- L. 299: Table 3: Please include F and p-value for column 1 in both figures 3 and 4.
R/. All these statistical parameters are given in table 3 to avoid excessive text in the figures. No changes applied.
- L. 304: This whole section needs rework and the last sub-figure in Figure 5 isn’t right based on the intercepts in Table 4.
R/. Changes applied. The figure was modified as well as the slope and intercepts.
- Discussion and Conclusions need to be revised accordingly based on the editing of Materials and methods and Results.
R/. Changes applied.
Reviewer 4 Report
The authors present a study on validation of satellite-derived NDVI observations and phenological metrics derived from MODIS and PROBA-V by comparison to observations from near-surface ground-based optical sensors set-up in four Tropical Dry Forest (TDF) sites located in the Americas. Validation of remote sensing products is essential to the proper use and interpretation of observations acquired from sensors aboard satellites for environmental monitoring and management.
General comments:
The authors have described the importance of the study in a broad context and described the experimental setup and processes carried out. The manuscript would benefit from:
Major English grammar revision. More detailed descriptions of the importance of the sites selected, e.g. from a regional environmental and socio-economic context, and therefore the main research gaps. A more detailed description of the near-surface optical phenological tower sensors and their data.Specific comments:
Please consider revising the title with respect to the proper use of title case. Consistency in terms used is imperative. Examples include: time-series vs. time series satellite-based vs. satellite-derived In the abstract, ‘spectroradiometric measurements’ is listed as a keyword but it is not used anywhere within the abstract or manuscript. Please go through the manuscript and provide acronyms of terms e.g. EBVs (line 41), NASA (line 45), and ESA (line 46). A detailed explanation or description of successional stages and types of forest cover is required in the introduction in order to connect to the site description (section 2.1), along with appropriate references, for the novice reader, and for a better understanding of the TDFs. Photos of the sensors as set up at any of the sites would add value to the description of the experimental setup. The description of the optical towers and sites in section 2.1 is confusing. In lines 90~91, is the height of 10.5m the canopy height or the height of the tower? What is the height of the tower at SRNP-EMSS? What is the canopy height at LC-SP? In line 115, is the instrument name and series (Apogee SQ-110) the same for both the silicon pyranometers and quantum sensors? In lines 116~117, the PAR range (400-700 nm) is captured within the incident and reflected shortwave solar radiation range. Please provide a detailed description of the near-surface ground-based sensors in terms of spectral resolution and bandwidth. In connection to comment 9, some studies have shown that spectral bandwidth and placement of the NIR band impact NDVI, especially when the NIR bandwidth is greater than 50nm, which is the case for PROBA-V. Please see, Teillet, P. M., Staenz, K., & William, D. J. (1997). Effects of spectral, spatial, and radiometric characteristics on remote sensing vegetation indices of forested regions. Remote Sensing of Environment, 61(1), 139-149. In this study, was this factored into the analysis as a source of differences? Observational tower footprint in this study is approximately 1.76 Ha (Line 120), while a MODIS pixel is approximately 6.25 Ha. Was this factored into the considerations accounting for the differences found in this study between satellite-derived and ground-based NDVI observations and phenological descriptors? If the difference in spatial footprint was considered, what was the effect? Same applies to the PROBA-V observations which has a pixel footprint of approximately 11.08 Ha. In line 119, please use the correct symbol for degrees. Please consider revising the grammar in Lines 151~153. Does it mean that PROBA-V observations were excluded due to unavailability of temporally coincident data at the PEMS-EMSS early and intermediate sites? In Figure 2, the LC-SP and PEMS-EMSS Late time-series plots appear to be visually incredibly similar. In addition, the dates for LC-SP do not match up to those provided in Table 1. Can supplementary data in the form of graphical tables be provided in order to verify that the data points are not the same? Prediction accuracy between the near-surface and satellite-derived forest phenophases is evaluated via regression (Lines 305~307). Some researchers have argued that the placement of variables, i.e. predicted (y-axis) versus observed data (x-axis) is erroneous. Please see, Piñeiro, G., Perelman, S., Guerschman, J. P., & Paruelo, J. M. (2008). How to evaluate models: observed vs. predicted or predicted vs. observed?. Ecological Modelling, 216(3-4), 316-322. In this study, as shown in Figure 5, which are the predicted values? By using the term ‘predict’ e.g. lines 18 and 73, it is implied that a model was applied or used to come up with the predicted values. Which model was used? In Figure 5, please consider using a different symbol for CBS or PEMS-EMSS in order to differentiate the two. In lines 347-349, clarification is necessary. The MODIS observations at all sites are in tandem with the optical tower time-series, as shown in Figure 2, and are much longer compared to PROBA-V. As such, if the analysis was modified in order to have observations from all sensors having the same length, would the same dissimilarities or differences be observed in the MODIS-tower observations? Granted, two of the sites, that is, the PEMS-EMSS Early and PEMS-EMSS Intermediate, would not be included in the analysis, but they are still not accounted for in the current analysis. Please provide a detailed discussion and references on the effect of anisotropy on NDVI with respect to the objectives of this study. As is, lines 372~375, 386~388 and 404~406 introduce the concept of anomalies, which are not analyzed in this study, and are in contradiction with lines 365~366.
Author Response
Reviewer 4
The authors present a study on validation of satellite-derived NDVI observations and phenological metrics derived from MODIS and PROBA-V by comparison to observations from near-surface ground-based optical sensors set-up in four Tropical Dry Forest (TDF) sites located in the Americas. Validation of remote sensing products is essential to the proper use and interpretation of observations acquired from sensors aboard satellites for environmental monitoring and management.
General comments:
The authors have described the importance of the study in a broad context and described the experimental setup and processes carried out. The manuscript would benefit from:
Major English grammar revision.
More detailed descriptions of the importance of the sites selected, e.g. from a regional environmental and socio-economic context, and therefore the main research gaps.
A more detailed description of the near-surface optical phenological tower sensors and their data.
Specific comments:
Please consider revising the title with respect to the proper use of title case.
R/. Changes applied.
Consistency in terms used is imperative. Examples include: time-series vs. time series satellite-based vs. satellite-derived.
R/. Changes applied. The consistency of these terms throughout the manuscript was reviewed.
In the abstract, ‘spectroradiometric measurements’ is listed as a keyword but it is not used anywhere within the abstract or manuscript.
R/. Changes applied. This keyword was replaced by ‘satellite observations’.
Please go through the manuscript and provide acronyms of terms e.g. EBVs (line 41), NASA (line 45), and ESA (line 46).
R/. The acronyms highlighted by the reviewer were not added since these words are used only once in the text; therefore, there is no need to use acronyms. No changes applied.
A detailed explanation or description of successional stages and types of forest cover is required in the introduction in order to connect to the site description (section 2.1), along with appropriate references, for the novice reader, and for a better understanding of the TDFs.
R/. Some changes applied. We added sentences with references in the introduction to describe the successional stages.
Photos of the sensors as set up at any of the sites would add value to the description of the experimental setup.
R/. Changes applied. A photo of the towers and the sensors were added into Figure 1.
The description of the optical towers and sites in section 2.1 is confusing. In lines 90~91, is the height of 10.5m the canopy height or the height of the tower? What is the height of the tower at SRNP-EMSS? What is the canopy height at LC-SP?
R/. Changes applied. The deployment of the towers and their heights was always based on the canopy height. By general rule, tower tends to be ~5-6 m above the forest canopy. In this new version, we added a column on Table 1 describing the tower height. Likewise, we rewrote some sentences in the description of sites.
In line 115, is the instrument name and series (Apogee SQ-110) the same for both the silicon pyranometers and quantum sensors? In lines 116~117, the PAR range (400-700 nm) is captured within the incident and reflected shortwave solar radiation range. Please provide a detailed description of the near-surface ground-based sensors in terms of spectral resolution and bandwidth.
R/. Changes applied. The PAR sensor is SQ-110 and the PYR is SP-100, this was corrected. Likewise, more information regarding the sensor features was added.
In connection to comment 9, some studies have shown that spectral bandwidth and placement of the NIR band impact NDVI, especially when the NIR bandwidth is greater than 50nm, which is the case for PROBA-V. Please see, Teillet, P. M., Staenz, K., & William, D. J. (1997). Effects of spectral, spatial, and radiometric characteristics on remote sensing vegetation indices of forested regions. Remote Sensing of Environment, 61(1), 139-149. In this study, was this factored into the analysis as a source of differences? Observational tower footprint in this study is approximately 1.76 Ha (Line 120), while a MODIS pixel is approximately 6.25 Ha. Was this factored into the considerations accounting for the differences found in this study between satellite-derived and ground-based NDVI observations and phenological descriptors? If the difference in spatial footprint was considered, what was the effect? Same applies to the PROBA-V observations which has a pixel footprint of approximately 11.08 Ha.
R/. The reviewer is correct - different bandwidths and their positions, as well as the spatial resolution may affect the NDVI values. This later has been demonstrated by Teillet et al. 1997 using hyperspectral imagery with high-spatial resolution. However, we believe that we should not compare hyperspectral imagery with broadband optical sensors. As Teillet et al. 1997 mentioned and we acknowledge, NDVI values are sensor dependent. Therefore, normalization values like those applied on section 2.6.2 are needed in order to decrease the effects imposed by the sensors on the NDVI observations. In terms of the footprint, unfortunately, there is no other way to do it, or at least we are not aware of how to do a factorization of the footprint from broadband optical sensors as a source of difference. As we mentioned to reviewer 1 above, it is unlikely and difficult to build a tower for a given height to cover a large pixel area, and it is unrealistic to build several towers around a given area to cover a large pixel. The selection of sites to build the towers was based on the homogeneity of forests; therefore, we can assume that the area cover by the near-surface sensor is representative of the area covered by the pixel. The latter is explicitly stated in the manuscript as assumptions to answer our questions. In addition, we acknowledge that differences between sensors may affect the NDVI observations and, for this reason, we avoided the use of the term ‘validation’ in our study. No changes applied.
In line 119, please use the correct symbol for degrees.
R/. Changes applied. The symbol was corrected.
Please consider revising the grammar in Lines 151~153.
R/. Changes applied.
Does it mean that PROBA-V observations were excluded due to unavailability of temporally coincident data at the PEMS-EMSS early and intermediate sites?
R/. Correct. Changes applied to improve the meaning.
In Figure 2, the LC-SP and PEMS-EMSS Late time-series plots appear to be visually incredibly similar. In addition, the dates for LC-SP do not match up to those provided in Table 1. Can supplementary data in the form of graphical tables be provided in order to verify that the data points are not the same?
R/. The reviewer is correct, those are similar because the figure is incorrect. In this new version, we correct the values of LC-SP. This error only impacted the graphical representation in this figure. Thanks for the observation.
Prediction accuracy between the near-surface and satellite-derived forest phenophases is evaluated via regression (Lines 305~307). Some researchers have argued that the placement of variables, i.e. predicted (y-axis) versus observed data (x-axis) is erroneous. Please see, Piñeiro, G., Perelman, S., Guerschman, J. P., & Paruelo, J. M. (2008). How to evaluate models: observed vs. predicted or predicted vs. observed?. Ecological Modelling, 216(3-4), 316-322. In this study, as shown in Figure 5, which are the predicted values? By using the term ‘predict’ e.g. lines 18 and 73, it is implied that a model was applied or used to come up with the predicted values. Which model was used?
R/. Changes applied. We understand the reviewer’s observation. In general, our aim is to compare how variable are the observations between satellite and near-surface sensors. In this new version, these comparisons were performed using standardized major axis and testing again slopes equal to 1; a perfect scenario of 1:1 relationship. Likewise, in this new version, the rationales behind x- y- axis was also addressed. We considered that the observed values are those from near-surface observations, while the predicted values are those from satellite sensors.
In Figure 5, please consider using a different symbol for CBS or PEMS-EMSS in order to differentiate the two.
R/. Changes applied. The figure was modified.
In lines 347-349, clarification is necessary.
R/. Changes applied.
The MODIS observations at all sites are in tandem with the optical tower time-series, as shown in Figure 2, and are much longer compared to PROBA-V. As such, if the analysis was modified in order to have observations from all sensors having the same length, would the same dissimilarities or differences be observed in the MODIS-tower observations? Granted, two of the sites, that is, the PEMS-EMSS Early and PEMS-EMSS Intermediate, would not be included in the analysis, but they are still not accounted for in the current analysis.
R/. The reviewer is correct, MODIS observations are much longer compared to PROBA-V, but at the same time, MODIS and PROBA-V were treated independently. This means that we ‘cut’ our observations twice in order to have the same length; one tower-MODIS and another one tower-PROBA-V. This implies, for example, that NDVI normalization values from tower observations at the same date could differ between MODIS or PROBA-V comparisons. In addition, PEMS-EMSS Early and Intermediate tower observations were not included in the comparisons against PROBA-V data from the satellite sensor was not available. The analysis does not account for these because we treated sites and sensors in different analysis. In order to avoid this misunderstanding, we rewrote some sections in methods.
Please provide a detailed discussion and references on the effect of anisotropy on NDVI with respect to the objectives of this study.
R/. Changes applied. More discussion and explanation about the anisotropy effects on NDVI were added.
As is, lines 372~375, 386~388 and 404~406 introduce the concept of anomalies, which are not analyzed in this study, and are in contradiction with lines 365~366.
R/. Changes applied. The concept of the anomalies was introduced. Likewise, previous lines 365-366 were rewritten.
Round 2
Reviewer 1 Report
The authors have sufficiently addressed all suggested changes from this reviewer. A minor spelling/grammar check is warranted, but other than that is acceptable.
Author Response
R/. We thank the reviewer for his/her comment. The English of the manuscript was extensively reviewed in this new version.
Reviewer 3 Report
Thanks for quick reply and modifications, and the manuscript got improved greatly. However, the two main concerns in the manuscript were still not addressed by the authors successfully.
1) Studies have been conducted in tropical ecosystems using traditional daily or MAIAC daily products in years (e.g., Hilker et al., 2012, 2015; Bi et al., 2016 and many more in Lyapustin et al., 2018) and confirmed the more accurate cloud mask and reduced noise in surface reflectance in MODIS MAIAC daily product. The MODIS daily products can be easily downloaded and processed as other traditional MODIS products. If the authors can add daily products in their comparison study, it would be a big plus for this manuscript. Even if the daily products cannot be added in this study, at least they need to be mentioned in the discussion or potential future work.
Bi, J., Myneni, R., Lyapustin, A., Wang, Y., Park, T., Chi, C., Yan, K., Knyazikhin, Y., 2016. Amazon Forests’ Response to Droughts: A Perspective from the MAIAC Product. Remote Sens-basel 8, 356.
Hilker, T., Lyapustin, A., Hall, F., Myneni, R., Knyazikhin, Y., Wang, Y., Tucker, C., Sellers, P., 2015. On the measurability of change in Amazon vegetation from MODIS. Remote Sens Environ 166, 233–242.
Hilker, T., Lyapustin, A., Tucker, C., Sellers, P., Hall, F., Wang, Y., 2012. Remote sensing of tropical ecosystems: Atmospheric correction and cloud masking matter. Remote Sens Environ 127, 370–384.
Lyapustin, A., Wang, Y., Korkin, S., Huang, D., 2018. MODIS Collection 6 MAIAC algorithm. Atmos Meas Tech 11, 5741–5765.
2) The authors emphasized that the near-surface observations were used as a point of reference. I still believe that the more reliable and effective direct comparison between two independent datasets should be conducted with original signals instead of smoothed signals. If the authors worried about the influence of extreme outliers, they can be filtered out. Only when in determining phenology using two independent datasets, cautious smoothing and gap-filling are needed.
Author Response
Thanks for quick reply and modifications, and the manuscript got improved greatly. However, the two main concerns in the manuscript were still not addressed by the authors successfully.
1) Studies have been conducted in tropical ecosystems using traditional daily or MAIAC daily products in years (e.g., Hilker et al., 2012, 2015; Bi et al., 2016 and many more in Lyapustin et al., 2018) and confirmed the more accurate cloud mask and reduced noise in surface reflectance in MODIS MAIAC daily product. The MODIS daily products can be easily downloaded and processed as other traditional MODIS products. If the authors can add daily products in their comparison study, it would be a big plus for this manuscript. Even if the daily products cannot be added in this study, at least they need to be mentioned in the discussion or potential future work.
Bi, J., Myneni, R., Lyapustin, A., Wang, Y., Park, T., Chi, C., Yan, K., Knyazikhin, Y., 2016. Amazon Forests’ Response to Droughts: A Perspective from the MAIAC Product. Remote Sens-basel 8, 356.
Hilker, T., Lyapustin, A., Hall, F., Myneni, R., Knyazikhin, Y., Wang, Y., Tucker, C., Sellers, P., 2015. On the measurability of change in Amazon vegetation from MODIS. Remote Sens Environ 166, 233–242.
Hilker, T., Lyapustin, A., Tucker, C., Sellers, P., Hall, F., Wang, Y., 2012. Remote sensing of tropical ecosystems: Atmospheric correction and cloud masking matter. Remote Sens Environ 127, 370–384.
Lyapustin, A., Wang, Y., Korkin, S., Huang, D., 2018. MODIS Collection 6 MAIAC algorithm. Atmos Meas Tech 11, 5741–5765.
R/. Thank you for your observation and comment. We appreciate your suggestion; however, we consider that the approach that you suggested may improve only the MODIS part of the manuscript, leaving the PROBA-V component with the same problem that you highlighted. Because of this, and the effects of the spatial resolution mentioned in our previous response, we still consider that our current methods are appropriate to answer our questions. Despite this, we agree with the reviewer that daily products comparison can be part of future work. This was mentioned at the end of the discussion in a new section (see section 4.4. starting on line 481).
2) The authors emphasized that the near-surface observations were used as a point of reference. I still believe that the more reliable and effective direct comparison between two independent datasets should be conducted with original signals instead of smoothed signals. If the authors worried about the influence of extreme outliers, they can be filtered out. Only when in determining phenology using two independent datasets, cautious smoothing and gap-filling are needed.
R/. Thank you for your comment. Despite the satellite NDVI products provide the ‘best pixels’ of a group of observations in a given time, they still have observations that seem to be anomalous. This can be seen in Figure S1 and S2. As such. we do not consider that the methods suggested by the reviewer are wrong, but neither are the methods used in our manuscript. We acknowledged the reviewer suggestion; at the end of the discussion, we mentioned this comment of direct comparisons as potential future work.
Reviewer 4 Report
The authors' efforts to improve the manuscript are well noted and appreciated. However, major issues to do with grammar and discussion of the results persist.
Further, the authors stress, in their response to the first review, that the main aim of the study is not validation but to explore the sources of variability in satellite-derived NDVI products, by comparison with near-surface observations. While this is appreciated, the results of the analysis carried out, as reported, are common knowledge, and the explanations on the sources of variability are riddled with contradictions.
Please kindly address the issues highlighted in the attached manuscript on a point-by-point basis.

Author Response
The authors' efforts to improve the manuscript are well noted and appreciated. However, major issues to do with grammar and discussion of the results persist.
Further, the authors stress, in their response to the first review, that the main aim of the study is not validation but to explore the sources of variability in satellite-derived NDVI products, by comparison with near-surface observations. While this is appreciated, the results of the analysis carried out, as reported, are common knowledge, and the explanations on the sources of variability are riddled with contradictions.
Please kindly address the issues highlighted in the attached manuscript on a point-by-point basis.
1) Line 23: Do you mean generalized linear models?
R/. Changes applied. It was corrected by ‘generalized’.
2) Line 58: Please consider revising the grammar.
R/. Changes applied.
3) Line 71: Please consider changing to “shed leaves”.
R/. Changes applied. Corrected as suggested by the reviewer.
4) Line 74: Use of an indefinite or non-specific reference, "these". Please consider revising.
R/. Changes applied. It was modified to ‘These elements of composition and structure’
5) Line 108: Them? Use of indefinite or non-specific reference. Please consider revising.
R/. Changes applied. It was modified to ‘varies between sites’.
6) Line 165: This was pointed out in the first review. Use of redundant language in ‘per each’.
Please consider revising.
R/. Changes applied. The redundant language of ‘per each’ was corrected in the manuscript.
7) Line 173: In this line and several other parts of the manuscript, please consider using 'daily'
instead of 1-day.
R/. Changes applied. ‘Daily’ is currently used in the manuscript.
8) Line 188: Interacts? Please consider revising the grammar.
R/. Changes applied. It was corrected to ‘interacts’.
9) Lines 200-202: Are the daily observations standardized? If not, given that the authors have
already highlighted the merits of using standardized observations for comparison, why use
non-standardized data?
R/. The non-standardized data was only used to extract phenometrics. These metrics are no affected if we use standardize or non-standardize data since they come from derivatives and the annual peaks and valleys of observations. The standardization affects the magnitude of the values, but not its trend. No changes applied.
10) Lines 203-204: As pointed out in the first review, this assumption is demonstrably invalid.
It can only be valid assuming that the land cover in the area covered by the optical sensor
is the same as that within a pixel of the satellite-derived NDVI product. This is made
apparent in Lines 205~207. Please consider revising the language in assumption (ii).
R/. Changes applied. The language of the assumption was corrected as the reviewer suggested.
11) Line 217, 227-229, 343, 346, 370, 378-379, 393-394, 397, 399, 410-411, 430-431: Please
consider revising the grammar
R/. Changes applied. The grammar of these lines was reviewed.
12) Line 256: Redundant since acronyms are already outlined in Lines 236-237.
R/. Changes applied. The acronyms on lines 236-237 were removed because we consider that it is better to describe these acronyms in the section where they belong.
13) Line 262: Phenometrics?
R/. Changes applied.
14) Line 264: Please kindly check the appropriate use of 'appropriated' and revise accordingly.
R/. Changes applied. It was corrected to ‘appropriate’.
15) Line 265: Spelling mistake. Is it meant to be “with”?
R/. Changes applied. It was corrected as the reviewer suggested.
16) Lines 379-380: The authors of the reference provided noted that "Planned adjustments to
Terra MODIS calibration for Collection 6 data reprocessing will largely eliminate this
negative bias in detection of NDVI trends". Since your study utilized MODIS collection 6
data, is it your conclusion that, based on your results, the effects of degradation in NDVI
inherent in pre-collection 6 are still prevalent?
R/. Thanks for your indirect suggestion. You are right; we do not have enough evidence to support this claim. Lines 380-386 were removed.
17) Line 388: Do you mean “temporally”? Please consider revising.
R/. Changes applied. It was corrected as the reviewer suggested.
18) Line 391: Why wasn't this implemented? Just because data for an extended time-series is
available, it doesn't mean that the entire time series has to be used, especially if this will
distort results or, time-series of the same length have been found to be better. This issue
was raised in the first review and has not been sufficiently addressed in this version of the
manuscript.
R/. It was not implemented since our aim is not to make comparisons between satellite sensors, our goal is to compare satellite sensors with tower observations (Lines 79-88). We added this sentence here in order to guide readers on how they could compare satellite sensors. No changes applied.
19) Lines 395-396: Doesn't this contradict lines 312-314? "Overall, although some
comparisons of NDVIDS between near-surface and satellite sensors reveal differences
between seasons and ENSO months, there is not a clear trend of higher or lower NDVIDS
values during the wet or dry season, or in a given ENSO month."
R/. Both lines 312-314 and 395-396 suggest that there are differences, but lines 312-314 suggest that there is no certainty of the direction of these differences. In other words, lines 312-314 are in the context that despite there are differences in some distributions, these distributions are not consistent between sites when looking to the trends of higher or lower values during the wet or the dry seasons. Lines 395-396 only suggest that there are differences. We added ‘some’ on line 396 to avoid misunderstandings.